# Policy Likelihood-based Query Sampling and Critic-Exploited Reset for Efficient Preference-based Reinforcement Learning

**Jongkook Heo[1], Jaehoon Kim[1], Young Jae Lee[2], Min Gu Kwak[3], Youngjoon Park[4], Seoung Bum Kim[1]***

[1]School of Industrial and Management Engineering, Korea University, [2]Samsung Electronics,
[3] Department of Health Information Management, University of Pittsburgh, [4]LG AI Research
[1]{hjkso1406, jhoon0418, sbkim1}@korea.ac.kr
[2]jae601.lee@samsung.com, [3]mik414@pitt.edu, [4]yj.park@lgresearch.ai

## Abstract

Preference-based reinforcement learning (PbRL) enables agent training without explicit reward design by leveraging human feedback. Although various query sampling strategies have been proposed to improve feedback efficiency, many fail to enhance performance because they select queries from outdated experiences with low likelihood under the current policy. Such queries may no longer represent the agent's evolving behavior patterns, reducing the informativeness of human feedback. To address this issue, we propose a policy likelihood-based query sampling and critic-exploited reset (PoLiCER). Our approach uses policy likelihood-based query sampling to ensure that queries remain aligned with the agent's evolving behavior. However, relying solely on policy-aligned sampling can result in overly localized guidance, leading to overestimation bias, as the model tends to overfit to early feedback experiences. To mitigate this, PoLiCER incorporates a dynamic resetting mechanism that selectively resets the reward estimator and its associated Q-function based on critic outputs. Experimental evaluation across diverse locomotion and robotic manipulation tasks demonstrates that PoLiCER consistently outperforms existing PbRL methods. Our code is available at https://github.com/JongKook-Heo/PoLiCER.

## 1 Introduction

Reinforcement learning from human feedback (RLHF) has emerged as a powerful paradigm for learning complex behaviors without manually designing reward functions, with applications across diverse domains (Kaufmann et al., 2024; Ouyang et al., 2022; Xu et al., 2023). In the context of Markov decision processes (MDPs), this approach is often referred to as preference-based reinforcement learning (PbRL) (Wirth et al., 2017). In PbRL, an annotator compares trajectories—sequences of state-action pairs—from the agent's history and indicates which is preferred (Christiano et al., 2017). Despite its advantages, PbRL faces two major challenges stemming from its dependence on sequentially collected human feedback. First, query-policy misalignment arises when sampled queries no longer represent the current policy's behavior, reducing the feedback effectiveness (Hu et al., 2024). Second, primacy bias causes the reward estimator to overweight early feedback, creating persistent biases that distort the reward model throughout training.

Hu et al. (2024) argued that query-policy misalignment occurs when queries are selected from long-past experiences that no longer represent the current policy. This reliance on outdated information diminishes feedback effectiveness and limits its ability to guide policy updates. To address this issue, Hu et al. (2024) introduced near on-policy sampling (NOS), which prioritizes queries from recent agent experiences based on the assumption that recency correlates with the current policy's behavior. While NOS proves effective in environments with repetitive behaviors, such as loco-

---

*Corresponding author.

motion tasks (Tassa et al., 2018; Tunyasuvunakool et al., 2020), where recent experiences remain representative, it struggles in goal-conditioned environments like Meta-World (Yu et al., 2020). In these settings, dynamic target locations make recent experiences unreliable indicators of current policy. Figure 1 illustrates this limitation by tracking policy likelihoods of replay buffer episodes during training across three categories: all past episodes (All), the most recent 30 episodes (Last 30), and the top 30 episodes ranked by current policy likelihood (Top 30). While recent episodes show better alignment with the current policy than the full history, a widening gap emerges between recent and high-likelihood episodes, demonstrating that temporal recency becomes an increasingly poor proxy for policy relevance. To overcome this limitation, we propose policy likelihood-based query sampling (PLS), which selects queries based on their likelihood under the current policy rather than temporal recency. Our method ensures that selected queries remain relevant throughout training by continuously adapting to the evolving policy.

The second critical challenge in PbRL is primacy bias—the tendency of neural networks to overfit to early data when trained on a growing dataset (Nikishin et al., 2022; Lyle et al., 2022; 2023; Kaufmann et al., 2024). In PbRL, the reward estimator suffers from this bias as it learns from continuously collected feedback. Early feedback, which is often trivial or ambiguous (Mu et al., 2025; Tu et al., 2025), can dominate the entire learning process. We found that this bias leads to reward overestimation: the reward estimator assigns excessively high values to early-stage favored state-actions, even when later feedback contradicts them. In particular, we identified that solely relying on PLS can highly localize the learning process, narrowing the policy distribution and further exacerbating overestimation. The biased rewards then propagate to the Q-function, reinforcing outdated behaviors and distorting policy optimization.

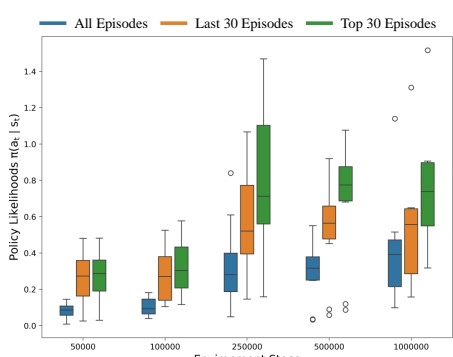

Figure 1: Policy likelihoods of replay buffer episodes in Meta-World Sweep Into (10,000/50), categorized into all past episodes (All), recent 30 episodes (Last 30), and top 30 episodes into current policy likelihood (Top 30). Temporal recency becomes an increasingly poor indicator of policy relevance as training progresses.

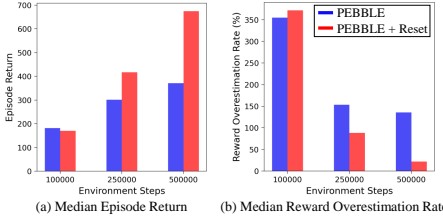

(a) Median Episode Return    (b) Median Reward Overestimation Rate

Figure 2: Performance comparison of PEBBLE and PEBBLE + Reset. (a) Average episode return and (b) reward overestimation rate on Walker Walk (100/10), showing median across ten runs. Primacy bias in the reward estimator leads to reward overestimation and performance degradation in PbRL.

To demonstrate that primacy bias in the reward estimator causes reward overestimation, we compared PEBBLE with PEBBLE + Reset—a variant that resets the reward estimator parameters at every feedback session, following standard primacy bias mitigating approaches (Ash & Adams, 2020; Nikishin et al., 2022; D'Oro et al., 2022). We quantified reward overestimation using the rate as $(\hat{R} - R)/R \times 100\%$, where $\hat{R}$ and $R$ denote the scaled estimated and true episode rewards, respectively. Figure 2 shows that periodic resetting substantially improves performance while reducing reward overestimation. Without resetting, PEBBLE consistently overestimates rewards, plateauing at approximately 150%. Furthermore, PEBBLE fails to distinguish fine-grained differences among trajectories, particularly for low-reward cases where reward overestimation is most severe (see Figure 14 in Appendix D.2). These results demonstrate how early, potentially misleading feedback can dominate the entire learning process.

While resetting the reward estimator at every feedback session proves effective at mitigating primacy bias-driven reward overestimation, this technique introduces two key challenges: (1) the high computational cost associated with frequent resets and (2) residual bias in the corresponding Q-function, which persists due to outdated reward estimates reinforced through temporal difference (TD) learning. To overcome these challenges, we propose critic-exploited reset (CER), a dynamic

resetting strategy that resets both the reward estimator and Q-function based on critic outputs. The critic output serves as an indicator of overestimation by approximating expected cumulative rewards (Van Hasselt et al., 2016; Nikishin et al., 2022; Chen et al., 2021). By strategically monitoring critic outputs, we trigger resets only when necessary, thus mitigating bias while maintaining computational efficiency.

In summary, we address two key challenges in PbRL: query-policy misalignment in query sampling and primacy bias in reward estimator learning. We propose policy likelihood-based query sampling and critic-exploited reset (PoLiCER), with each component targeting a specific challenge. First, PLS ensures queries remain aligned with the current policy by selecting trajectories based on their likelihood rather than temporal recency. Second, CER mitigates primacy bias through dynamic resetting of both the reward estimator and Q-function based on critic outputs, providing computational efficiency while addressing residual bias. We evaluated PoLiCER on complex continuous control tasks in the DeepMind Control Suite (DMControl) (Tassa et al., 2018; Tunyasuvunakool et al., 2020) and Meta-World (Yu et al., 2020). Comprehensive experimental results demonstrate that PoLiCER significantly improves performance when integrated with PEBBLE (Lee et al., 2021b) and outperforms other PbRL methods with complementary components.

## 2 RELATED WORKS

**Preference-based Reinforcement Learning.** Christiano et al. (2017) established the foundation for training deep RL agents using human preferences (Mnih et al., 2016; Schulman et al., 2017). Lee et al. (2021b) built upon this work, incorporating soft actor-critic (SAC) (Haarnoja et al., 2018), unsupervised pre-training, and reward relabeling to enhance sample efficiency. Subsequent studies have pursued various directions to improve feedback efficiency: SURF (Park et al., 2022) leveraged unlabeled queries through semi-supervised reward learning, RUNE (Liang et al., 2022) used reward uncertainty for exploration, and MRN (Liu et al., 2022) applied bi-level optimization to reward learning. Other advances include robust learning under noisy preferences (Cheng et al., 2024; Huang et al., 2025), dynamics modeling (Metcalf et al., 2023), preference credit assignment (Verma & Metcalf, 2024), and vision-language model (VLM)-based reward design (Wang et al., 2024; Tu et al., 2025). Among these approaches, our work is most closely related to QPA (Hu et al., 2024), which identified query-policy misalignment. While QPA relies on recency for query sampling, we showed that recent data selection inadequately captures policy relevance. Instead, we propose sampling queries based on policy likelihoods for more effective and policy-relevant query selection. DUO (Feng et al., 2025) similarly leverages policy likelihood through rectified Z-scores, whereas we employ ranks of trajectory likelihoods without buffer normalization. Complementary to query selection methods, PPE (Zhu et al., 2024) tackled query-policy misalignment through an exploration strategy that adjusts the behavior policy near the current policy to expand data coverage in policy-relevant regions. In contrast, we directly address query-policy misalignment by identifying policy-relevant queries from the replay buffer based on their likelihoods, without requiring modifications to the exploration strategy or additional behavior policy.

**Primacy Bias and Loss of Generalization Ability in Neural Networks.** PbRL poses unique challenge beyond standard RL: periodic reward model updates create non-stationary reward signals, while high reliance on approximate rewards risks overoptimization (Kaufmann et al., 2024). These issues are compounded by primacy bias, a key challenge in deep RL identified by Nikishin et al. (2022), who proposed periodic resets as a mitigation strategy. D'Oro et al. (2022) later demonstrated that more frequent resets enable higher replay ratios, improving sample efficiency. More broadly, primacy bias relates to a wider set of neural network degradation phenomena, including *capacity loss*—the gradual loss of network adaptability to new targets (Lyle et al., 2022)—and *plasticity loss*—the inability to overwrite prior predictions when input-output relationships change (Lyle et al., 2023). These issues are particularly severe in RL due to its non-stationary nature (Igl et al., 2021). Proposed solutions to counteract these degradations include feature space regularization (Lyle et al., 2022), loss landscape flattening (Lyle et al., 2023; Lee et al., 2023; Foret et al., 2021), plasticity injection (Nikishin et al., 2023), and resetting inactive neurons (Nikishin et al., 2022; D'Oro et al., 2022; Sokar et al., 2023; Xu et al., 2024). Notably, Nauman et al. (2024) identified resetting as the most effective and robust intervention. Building upon these insights, we develop an adaptive resetting mechanism tailored for PbRL.

## 3 REWARD ESTIMATOR LEARNING WITH PREFERENCE DATASET

PbRL trains agents to align with human intent using human feedback. This approach replaces traditional reward functions with a learned reward estimator that captures human preferences. The process begins by collecting pairs of agent behavior trajectories and corresponding preference labels, where human annotators indicate which trajectory they prefer. The reward estimator is then trained to assign higher rewards to preferred trajectories and lower rewards to non-preferred ones. This learned reward function guides the agent's policy optimization in standard RL.

At each feedback session, trajectory pairs are sampled for human evaluation. A trajectory is defined as a fixed-length sequence of state-action pairs, $\sigma = \{(s_0, a_0), (s_1, a_1), \ldots, (s_{H-1}, a_{H-1})\} \in (S \times A)^H$, where $H$ is the trajectory length. For each pair $(\sigma^0, \sigma^1)$, the human teacher provides a binary preference label $y$, typically in one of three forms: $(1, 0)$ when preferring $\sigma^0$, $(0, 1)$ when preferring $\sigma^1$, or $(0.5, 0.5)$ to indicate equal preference. Each preference instance $(\sigma^0, \sigma^1, y)$ is accumulated in a dataset $D = \{(\sigma^0, \sigma^1, y)_i\}_{i=1}^N$, where $N$ represents the total feedback collected.

Using this dataset $D$, the reward estimator $\hat{r}_\psi : S \times A \to \mathbb{R}$ is trained to align predicted rewards with human preferences. The preference probability is modeled using the Bradley-Terry model (Bradley & Terry, 1952):

$$P_\psi(\sigma^1 \succ \sigma^0) = \frac{\exp(\sum_{(s_t, a_t) \in \sigma^1} \hat{r}_\psi(s_t, a_t))}{\exp(\sum_{(s_t, a_t) \in \sigma^1} \hat{r}_\psi(s_t, a_t)) + \exp(\sum_{(s_t, a_t) \in \sigma^0} \hat{r}_\psi(s_t, a_t))}. \tag{1}$$

Herein, $\sigma^1 \succ \sigma^0$ denotes that $\sigma^1$ is preferred over $\sigma^0$. $\hat{r}_\psi$ is trained to minimize the binary cross-entropy (BCE) loss between true preference labels and predicted preference probabilities:

$$L_\psi = -\mathbb{E}_{(\sigma^0, \sigma^1, y) \sim D}[y(0) \log P_\psi(\sigma^0 \succ \sigma^1) + y(1) \log P_\psi(\sigma^1 \succ \sigma^0)], \tag{2}$$

where $y(i)$ indicates the $i$-th component of $y$. This framework seamlessly integrates with various RL algorithms (Mnih et al., 2016; Schulman et al., 2017; Haarnoja et al., 2018; Yarats et al., 2022), eliminating the need for manual reward design.

## 4 PROPOSED METHOD

In this section, we present PoLiCER, which consists of two key components: (1) policy likelihood-based query sampling (PLS) and (2) critic-exploited reset (CER). PLS addresses query-policy misalignment by selecting informative queries based on their likelihoods under the current policy. CER mitigates overestimation caused by primacy bias by strategically resetting both the reward estimator and its corresponding Q-function as the feedback dataset accumulates.

### 4.1 POLICY LIKELIHOOD-BASED QUERY SAMPLING

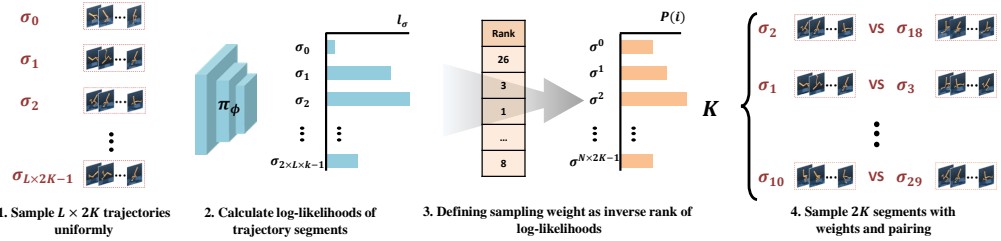

Figure 3: Overall framework of PLS. Log-likelihoods of trajectories are used for sampling informative queries relevant to the current policy. The inverse rank of log-likelihoods serves as probability weights, and the hyperparameter $\alpha$ controls the sharpness of the distribution.

Unlike previous approaches that use recency as a proxy for policy relevance, our method directly quantifies how well each trajectory represents the current policy by computing its log-likelihood,

ensuring alignment regardless of when the data was collected. Figure 3 illustrates the overall process. At each feedback session, we begin by uniformly sampling $2 \times L \times K$ trajectories, where $K$ is the number of queries to extract and $L$ is a scaling factor, following prior weighted sampling approaches (Christiano et al., 2017; Lee et al., 2021a;b). We then compute the average log-likelihood of each trajectory under the current policy $\pi_\phi$ as follows:

$$l_i = \frac{1}{H} \sum_{(s_t, a_t) \sim \sigma_i} \log \pi_\phi(a_t \mid s_t). \tag{3}$$

The log-likelihood $l_i$ is proportional to the probability of trajectory $\sigma_i$ under policy $\pi_\phi$, with higher values indicate trajectories that better reflect current behavior of the agent. However, in continuous control tasks, log-likelihoods can be highly sensitive to outliers because of the unbounded nature of probability densities. To ensure robustness, we use the inverse rank of log-likelihoods as the sampling weight $w_i$ rather than raw values (Schaul et al., 2016):

$$p(i) = w_i^\alpha / \sum_j w_j^\alpha. \tag{4}$$

Here, $\alpha$ controls the sharpness of the sampling distribution: lower values yield more uniform sampling, while higher values favor high-likelihood trajectories. Finally, preference queries are generated by randomly pairing two sampled trajectories.

Compared to disagreement sampling (DS) (Christiano et al., 2017; Lee et al., 2021a), our method is computationally efficient, requiring only $2 \times L \times K$ forward passes, while DS incurs a cost that is $N$ times higher (where $N$ is the number of reward estimators in an ensemble). The pseudo code for PLS is presented in Algorithm 1 in Appendix B. Figure 13 in Appendix D.1 confirms that PLS selects queries more relevant to the current policy than NOS, with no practical increase in training time.

## 4.2 CRITIC-EXPLOITED RESET

As the feedback dataset grows, the reward estimator overfits to early feedback due to primacy bias, amplifying reward differences through the BCE loss in Equation 2 (Heo et al., 2025). While underestimation is less problematic since low-reward state-action pairs are naturally avoided (Fujimoto et al., 2018), overestimation proves more detrimental by encouraging suboptimal behaviors. Although periodic resetting helps, naive approaches suffer from computational inefficiency and residual Q-function bias. To address these challenges, we propose CER, a dynamic resetting strategy that leverages critic outputs as indicators of reward overestimation. Rather than resetting at fixed interval, CER monitors critic outputs against an adaptive threshold to determine when intervention is necessary. We design the threshold to increase monotonically throughout training for two critical purposes: first, it accommodates genuine policy improvement by allowing higher Q-values as the policy optimizes; second, it reduces unnecessary resets over time, avoiding excessive training disruption.

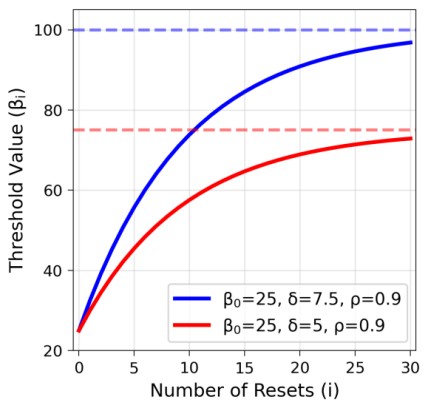

Figure 4: Visualization of CER threshold $\beta$ for different step sizes. Thresholds grow monotonically and saturate at upper bounds determined by $\delta$. Lower $\delta$ compensates for task complexity by maintaining tighter overestimation control.

Specifically, CER continuously tracks the maximum critic output $Q_{\max}$ observed across all gradient steps since the last reset. At each feedback session, if $Q_{\max}$ exceeds the current threshold $\beta_i$, CER resets both the reward estimator $r_\psi$ and Q-function $Q_\theta$ before updating the reward function. In PbRL, rewards are typically bounded to $[-1, 1]$ via $\tanh$ activation, constraining critic outputs to $|\frac{r}{1-\gamma}| = 100$ (with discount factor $\gamma = 0.99$). This bounded nature enables task-agnostic monitoring with consistent scale. We achieve monotonic threshold growth while ensuring a clear upper-bound through an exponential decay schedule:

$$\beta_{i+1} = \beta_i + \delta \times \rho^i, \tag{5}$$

where $\delta > 0$ is the initial step size, $\rho \in (0,1)$ controls decay rate, and $i$ denotes the number of resets performed. Here, $\delta$ directly regulates threshold growth based on task complexity (e.g., action dimension). This exponential decay allows the threshold to grow monotonically while gradually saturating at $\beta_0 + \frac{\delta}{1-\rho}$. If the threshold is not exceeded, training proceeds normally with the existing parameters. Figure 4 illustrates this adaptive threshold mechanism for different $\delta$ values.

After each reset, we increase the replay ratio to accelerate adaptation to updated reward estimates. Inspired by adaptive replay ratio scaling (Ma et al., 2024), we maintain low replay ratios during early training to preserve plasticity, then gradually increase them later for sample efficiency. Once feedback collection ends and reward learning completes, resets cease and the replay ratio returns to one, as further overrides become unnecessary. Additional details are provided in Algorithm 2 in Appendix B. Empirically, CER achieves higher returns and lower reward overestimation than naively resetting the reward estimator at every feedback session (see Figure 15 in Appendix D.2).

CER is grounded in three key theoretical principles and empirical findings: (1) resets eliminate accumulated primacy bias and corresponding overestimation (Nauman et al., 2024), (2) critic outputs serve as indirect indicators of overestimation (Van Hasselt et al., 2016; Fujimoto et al., 2018; Nikishin et al., 2022), and (3) valid policy improvements increase Q-values monotonically (Sutton et al., 1998). Combined with bounded rewards, these principles justify our monotonic threshold schedule with asymptotic saturation: initial low thresholds address early overestimation, while gradually increasing thresholds distinguish genuine policy improvements from bias-induced anomalies. Our empirical sensitivity analysis (Appendix D.7) further validates the robustness of CER across hyperparameters. We formalize CER's theoretical advantage through value approximation error bounds, adapting the analysis from Hu et al. (2024).

**Definition 1.** *Let $Q_r^\pi$ be the Q-function of the current stochastic policy $\pi$ under the true reward $r$. For a distribution $\mu$, define the distribution-dependent norm $\|f\|_\mu := \mathbb{E}_{x \sim \mu}[|f(x)|]$, and let $\mathcal{F}$ denote the function approximator class. The intrinsic function approximation error is defined as $\alpha_r^\pi := \inf_{f \in \mathcal{F}} \|Q_r^\pi - f\|_{d^\pi}$.*

**Lemma 1.** *Assume the reward overestimation error satisfies $\|\hat{r}_\psi - r\|_{d^\pi} \leq \varepsilon$. Further assume the learned Q-function estimator $\hat{Q}_{\hat{r}_\psi}^\pi$ is the empirical risk minimizer onto $\mathcal{F}$, so that $\hat{Q}_{\hat{r}_\psi}^\pi \in \arg\min_{f \in \mathcal{F}} \|Q_{\hat{r}_\psi}^\pi - f\|_{d^\pi}$. Then $\|Q_r^\pi - \hat{Q}_{\hat{r}_\psi}^\pi\|_{d^\pi} \leq \alpha_r^\pi + \frac{2\varepsilon}{1-\gamma}$ (Proof in Appendix A).*

**Lemma 2.** *Suppose CER reduces the reward overestimation error bound from $\varepsilon$ to $\varepsilon_C$, where $\delta = \varepsilon - \varepsilon_C > 0$ quantifies this improvement. Then CER tightens the value approximation error bound, guaranteeing an improvement of $\frac{2\delta}{1-\gamma}$ (Proof follows directly from Lemma 1 by substituting $\varepsilon_C$ for $\varepsilon$ and subtracting the bounds).*

These results quantify CER's advantage: reducing reward overestimation by $\delta$ yields a Q-function approximation error reduction of $\frac{2\delta}{1-\gamma}$. This provides theoretical justification for CER's design and offers a principled method to quantify the benefits of mitigating primacy bias.

## 5 EXPERIMENTS

### 5.1 SET UP

PoLiCER is compatible with any PbRL algorithm. In our experiments, we implemented it on PEBBLE (Lee et al., 2021b) as the baseline. For PLS, we set the temperature parameter $\alpha = 1$ across all main experiments. For CER, the initial threshold was set to $\beta_0 = 25$ with a decay rate $\rho = 0.9$. The initial step size was defined as $\delta = \frac{30}{|A|}$, to compensate for task complexity using action dimension $|A|$ (i.e., $\delta = 7.5$ for all Meta-World tasks and $\delta \approx 1.43$ for DMControl Humanoid). To improve feedback efficiency, we applied multiple temporal data augmentation (TA) with ratio $\tau = 20$, following prior works (Hu et al., 2024; Park et al., 2022). The pseudo code of PoLiCER is presented in Algorithm 3 in Appendix B. We evaluated our proposed method on three locomotion tasks from DMControl (Tassa et al., 2018; Tunyasuvunakool et al., 2020) and four robotic manipulation tasks from Meta-World (Yu et al., 2020). Following prior studies, we used a synthetic human annotator

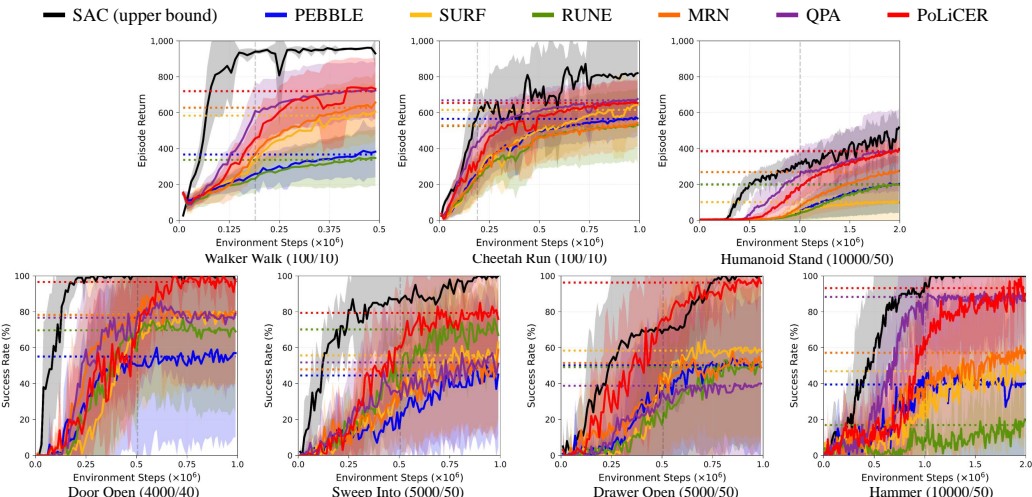

Figure 5: Learning curves using vector-based state inputs. First row: performance results for locomotion tasks (Walker Walk, Cheetah Run, Humanoid Stand) in DMControl. Second row: results for robotic tasks (Door Open, Sweep Into, Drawer Open, Hammer) in Meta-World.

that provides preference feedback based on ground truth rewards in B-Pref benchmark (Lee et al., 2021a). Comprehensive experimental details are provided in Appendix C.

## 5.2 VECTOR-BASED CONTROL WITH SYNTHETIC PREFERENCES

For a comprehensive comparison, we included SAC (Haarnoja et al., 2018) trained with ground truth rewards, along with other representative PbRL methods: SURF (Park et al., 2022), RUNE (Liang et al., 2022), MRN (Liu et al., 2022), and QPA (Hu et al., 2024). Each algorithm was evaluated at every 10,000 steps with ten evaluation episodes per task. In DMControl, we reported average episode returns, while in Meta-World, we used average success rates. Figure 5 presents the learning curves for all methods using vector-based state inputs in DMControl and Meta-World. Solid lines and shaded regions indicate the mean and standard deviation across ten runs. Vertical dashed gray lines mark the final feedback step, and horizontal dotted lines represent the final performance, computed by averaging the last ten evaluation scores for each method.

In DMControl, both PoLiCER and QPA showed significant performance improvements over PEBBLE. While their average performances were comparable, PoLiCER exhibited superior robustness with lower variance, as reflected in the 95% bootstrapped confidence intervals (see Figure 16 in Appendix D.3). In contrast, in Meta-World environments, most PbRL methods showed high variance and poor average success rate. PoLiCER, however, achieved over 80% success with consistently lower variance. The limited improvements of QPA in Meta-World suggest that its policy-aligned sampling strategy is ineffective in goal-conditioned tasks. Notably, in Drawer Open, none of the other methods exceeded a 60% success rate after feedback sessions ended, with nearly half of their runs resulting in 0% success. This indicates a failure to learn a generalized reward function that aligns feedback, which highlights the need for a deeper analysis of PoLiCER's individual components, as discussed in Section 5.3.

## 5.3 ABLATION STUDY - COMPONENT ANALYSIS

We evaluated the impact of each PoLiCER component on overall performance by incrementally incorporating PLS, CER, and TA. Figure 6 shows the learning curves and critic outputs across three representative tasks: Walker Walk, Door Open, and Drawer Open, to assess the effectiveness of each component. As shown in Figure 6 (a), both PLS and CER independently led to substantial performance gains. Although their combination was not always complementary, especially in Drawer Open, it remained generally effective in most environments. TA offered modest improvements: it slightly increased average scores in Door Open and reduced variance in Drawer Open, showing better generalization by the reward estimator. However, Figure 6 (b) reveals that PLS can induce critic overestimation due to its localized policy guidance, a phenomenon also observed in QPA (see Figure

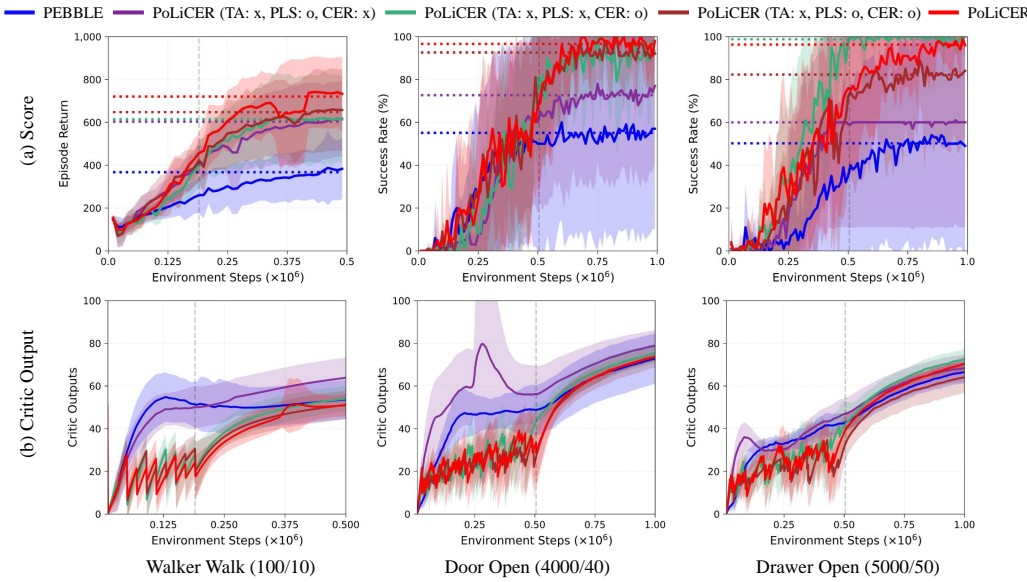

Figure 6: Contribution of each component in PoLiCER across Walker Walk, Door Open, and Drawer Open. In both rows, solid lines and shaded areas represent the average and standard deviation. (a) Learning curves showing average episode returns or success rates. (b) Critic outputs during training, used to indirectly assess overestimation.

17 in Appendix D.4). While this guidance improved average performance over PEBBLE, it can also reinforce overestimation, ultimately leading to suboptimal policies. CER helps mitigate this effect, enabling the agent to explore a broader policy space while still leveraging policy-aligned sampling.

Notably, in Drawer Open, CER alone achieved the highest success rate of approximately 100%, outperforming other combinations of components and PbRL methods. This result can be attributed to the misalignment between task success and rewards, particularly during the early and mid-training phases as shown in Figure 7. In these phases, the true reward distributions of successful and failed trajectories exhibit significant overlap, indicating that the true reward function does not consistently reflect actual task completion (see rendered examples in Appendix D.4). This discrepancy can introduce noisy preferences in early training stages, which may misguide both policy and

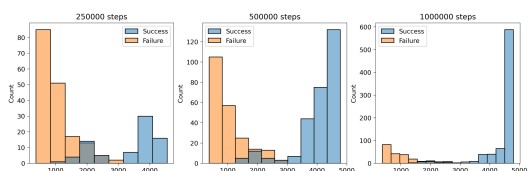

Figure 7: True reward distributions for successful and failed trajectories in Drawer Open under Po-LiCER at different training stages. Early and mid-training show low-reward trajectories even among successful attempts, revealing misalignment between task success and rewards.

query sampler. As a result, agents relying solely on PLS showed high performance variance and limited improvement in Drawer Open. In contrast, CER effectively reduced the influence of these early misaligned preferences, preventing the agent from being trapped in biased reward estimations and enabling more stable and consistent learning.

## 5.4 PIXEL-BASED CONTROL WITH SYNTHETIC PREFERENCES

We expanded our experiments to pixel-based state inputs using synthetic preference annotators. Following Park et al. (2022), we adopted DrQ-v2 (Yarats et al., 2022) as the backbone RL algorithm and included its variant of PEBBLE, SURF, and QPA (see the details in Appendix C.2). Our method achieved higher performance compared to other methods, as shown in Figure 8. Specifically, QPA failed to improve performance compared to the baseline (PEBBLE) in Cheetah Run and Window Open. This result suggests that the reliance of QPA on recent experiences for query sampling and

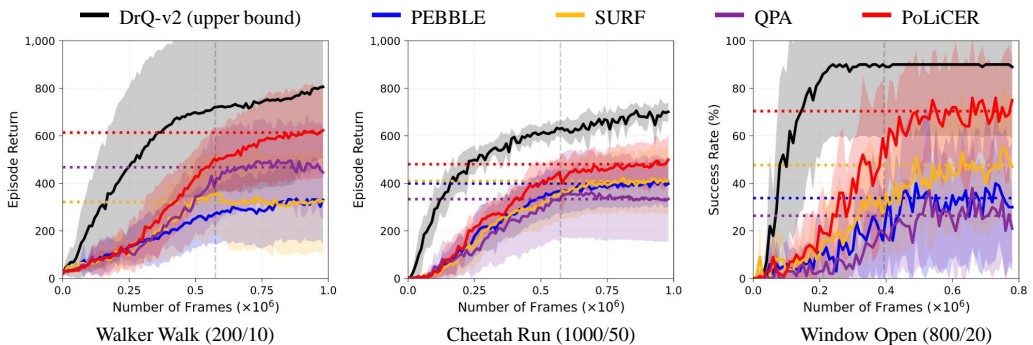

Figure 8: Learning curves on two locomotion tasks (Walker Walk and Cheetah Run), and one robotic manipulation task (Window Open) using pixel-based state inputs.

policy updates may be less effective in high-dimensional pixel-based environments, where visual complexity weakens the link between recency and relevance. In contrast, PoLiCER achieved approximately 80% success rate with low variance in Window Open, doubling the performance of PEBBLE. These findings demonstrate that prioritizing policy relevance over recency becomes increasingly crucial in complex visual domains, and our strategic reset approach provides a clear advantage for robust and stable preference learning across diverse perceptual challenges. While PoLiCER requires moderately longer training time than baselines due to CER's increased replay ratio after resets, this computational overhead is modest relative to the inherent rendering costs of pixel-based and is offset by substantial performance improvements (see Appendix D.5 for detailed computational analysis).

## 5.5 HUMAN-IN-THE-LOOP EXPERIMENT

While synthetic annotators in the B-Pref benchmark (Lee et al., 2021a) provide a fair comparison of PbRL methods, they may yield different trends and interpretations from human labels (Hu et al., 2024; Kim et al., 2023). As shown in Section 5.3, synthetic preferences do not always align with intended behaviors, which can lead to misleading conclusions. To ensure more realistic evaluations, we conducted human-in-the-loop experiments using ten volunteers on Walker Walk and Window Open for both PEBBLE and PoLiCER. Following prior works (Hu et al., 2024; Heo et al., 2025), we included an equal preference option $y = (0.5, 0.5)$ to account for human inconsistencies. Detailed experimental procedures for these tasks are provided in Appendix D.6.

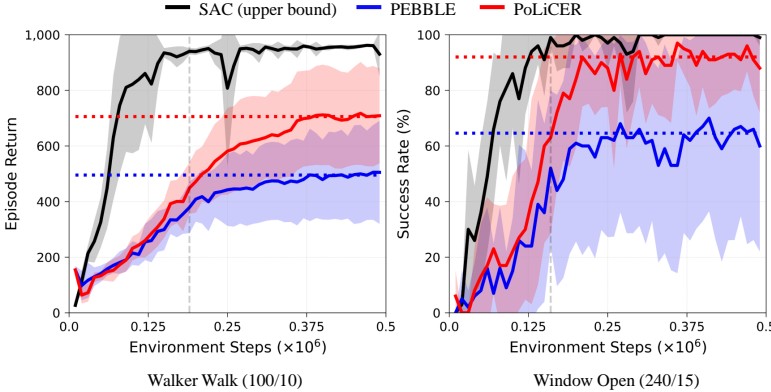

Figure 9: Learning curves on Walker Walk and Window Open using vector-based state inputs with human annotators.

Figure 9 shows that PoLiCER improved feedback efficiency over PEBBLE. Although both agents performed similarly during the feedback collection phase, PoLiCER achieved significantly higher

ground truth rewards and success rates as training progressed. We examined how the two methods produced qualitatively different learned behaviors. On Walker Walk, Figure 10 reveals striking differences: the PEBBLE-trained agent primarily moved by crawling on its knees with a hunched posture, failing to develop natural walking behavior. In contrast, the PoLiCER-trained agent successfully learned coordinated bipedal locomotion with alternating leg movements and maintained an upright torso throughout motion. This behavioral difference suggests that PoLiCER more effectively captured and incorporated human preferences regarding natural walking motion, translating feedback into more naturalistic and biomechanically appropriate locomotion patterns. These human-in-the-loop results provide critical validation that PoLiCER effectively narrows the gap between theoretical preference learning and practical implementation challenges, suggesting a promising approach for preference-based learning in complex interactive scenarios.

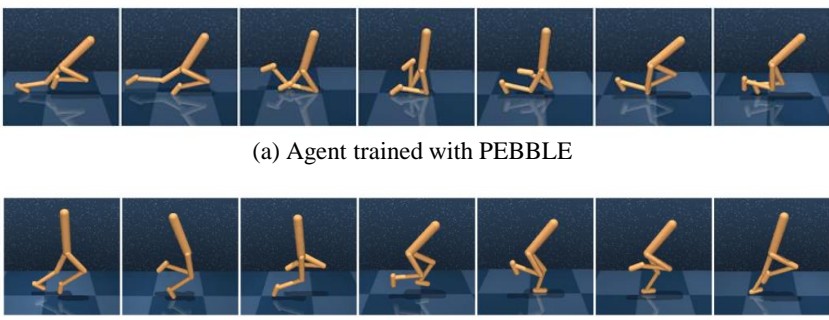

(a) Agent trained with PEBBLE

(b) Agent trained with PoLiCER

Figure 10: Comparison of final behaviors learned in Walker Walk with human feedback. (a) PEBBLE agent resorted to crawling with poor posture, while (b) PoLiCER agent demonstrated coordinated bipedal walking with proper balance and form.

## 6 CONCLUSION

In this study, we introduced PoLiCER, a novel approach that addresses two key challenges in PbRL: (1) query-policy misalignment in query sampling and (2) reward overestimation caused by primacy bias in reward learning. Our solution combines two complementary components: PLS ensures selected queries better reflect current policy behavior, while CER dynamically resets both the reward estimator and Q-function to mitigate primacy bias. Together, these components create a synergistic effect—PLS provides more relevant feedback signals, and CER prevents bias accumulation when processing this feedback. Our theoretical analysis proves that CER's resetting strategy delivers measurable improvements: for every unit of reward overestimation reduced, the Q-function approximation error improves by twice that amount, scaled by the discount factor. Extensive evaluations on DMControl and Meta-World benchmarks demonstrated that PoLiCER significantly outperformed existing PbRL methods, achieving nearly 100% success rate in Drawer Open where other methods struggled to exceed 60%. These results highlight the effectiveness of our integrated approach, which aligns query selection with policy behavior while maintaining stable learning through strategic resets.

## REPRODUCIBILITY STATEMENT

To ensure the reproducibility of our work, we described the implementation details in Appendix C, including all hyperparameters, network architecture, and training procedures (Table 1–12). The complete algorithms are provided in Algorithms 1–3 in Appendix B. Source code is publicly available at `https://github.com/JongKook-Heo/PoLiCER`.

## ACKNOWLEDGEMENTS

This research was supported by Culture, Sports and Tourism R&D Program through the Korea Creative Content Agency grant funded by the Ministry of Culture, Sports and Tourism in 2024(Project Name: Development of AI-based large-scale automatic game verification technology to improve game production verification efficiency for small and medium-sized game companies, Project Number: RS-2024-00393500, Contribution Rate: 100%)

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

## A  PROOFS

*Proof of Lemma 1.* Using the Bellman operator and Jensen's inequality, we have

$$
\begin{aligned}
\|Q^\pi_{\hat{r}_\psi} - Q^\pi_r\|_{d^\pi} &= \mathbb{E}_{(s,a)\sim d^\pi}\big| Q^\pi_{\hat{r}_\psi}(s,a) - Q^\pi_r(s,a)\big| \\
&\leq \mathbb{E}_{(s,a)\sim d^\pi}\Big|\hat{r}_\psi(s,a) - r(s,a) \\
&\qquad\qquad + \gamma\,\mathbb{E}_{s'\sim T(\cdot|s,a),\,a'\sim\pi(\cdot|s')}\big[Q^\pi_{\hat{r}_\psi}(s',a') - Q^\pi_r(s',a')\big]\Big| \\
&\leq \mathbb{E}_{(s,a)\sim d^\pi}\big|\hat{r}_\psi(s,a) - r(s,a)\big| + \gamma\,\mathbb{E}_{s'\sim T,\,a'\sim\pi}\big|Q^\pi_{\hat{r}_\psi}(s',a') - Q^\pi_r(s',a')\big| \\
&\leq \varepsilon + \gamma\,\|Q^\pi_{\hat{r}_\psi} - Q^\pi_r\|_{d^\pi}.
\end{aligned}
$$

Hence,

$$
(1-\gamma)\,\|Q^\pi_{\hat{r}_\psi} - Q^\pi_r\|_{d^\pi} \leq \varepsilon \quad\Rightarrow\quad \|Q^\pi_{\hat{r}_\psi} - Q^\pi_r\|_{d^\pi} \leq \frac{\varepsilon}{1-\gamma}.
$$

Let $f^\star \in \arg\min_{f\in\mathcal{F}} \|Q^\pi_r - f\|_{d^\pi}$ so that $\|Q^\pi_r - f^\star\|_{d^\pi} = \alpha^\pi_r$. Then, by the triangle inequality,

$$
\begin{aligned}
\alpha^\pi_{\hat{r}_\psi} = \inf_{f\in\mathcal{F}} \|Q^\pi_{\hat{r}_\psi} - f\|_{d^\pi} &\leq \|Q^\pi_{\hat{r}_\psi} - f^\star\|_{d^\pi} \\
&\leq \|Q^\pi_{\hat{r}_\psi} - Q^\pi_r\|_{d^\pi} + \|Q^\pi_r - f^\star\|_{d^\pi} \\
&\leq \tfrac{\varepsilon}{1-\gamma} + \alpha^\pi_r.
\end{aligned}
$$

Finally, decompose total error and use triangle inequality again:

$$
\begin{aligned}
\|Q^\pi_r - \hat{Q}^\pi_{\hat{r}_\psi}\|_{d^\pi} &\leq \|Q^\pi_r - Q^\pi_{\hat{r}_\psi}\|_{d^\pi} + \|Q^\pi_{\hat{r}_\psi} - \hat{Q}^\pi_{\hat{r}_\psi}\|_{d^\pi} \\
&= \|Q^\pi_r - Q^\pi_{\hat{r}_\psi}\|_{d^\pi} + \inf_{f\in\mathcal{F}} \|Q^\pi_{\hat{r}_\psi} - f\|_{d^\pi} \\
&\leq \tfrac{\varepsilon}{1-\gamma} + \left(\tfrac{\varepsilon}{1-\gamma} + \alpha^\pi_r\right) = \alpha^\pi_r + \tfrac{2\varepsilon}{1-\gamma}.
\end{aligned}
$$

**Remark 1.** In our proof we used the relation $\alpha^\pi_{\hat{r}_\psi} \leq \frac{\varepsilon}{1-\gamma} + \alpha^\pi_r$, which leads to the looser bound $\alpha^\pi_r + \frac{2\varepsilon}{1-\gamma}$. Under the more restrictive assumption that the function approximation error is independent of the reward function (i.e., $\alpha^\pi_{\hat{r}_\psi} = \alpha^\pi_r$), the total error bound reduces to $\alpha^\pi_r + \frac{\varepsilon}{1-\gamma}$, which coincides with the result in Hu et al. (2024).

## B  ALGORITHM DETAILS

In this section, we present the detailed procedures of PoLiCER, which consists of two components: (1) PLS and (2) CER. The complete implementation with PEBBLE is described in Algorithm 3.

---

**Algorithm 1** PLS

1: **Require:** Hyperparameters: temperature parameter $\alpha$, the number of queries per session $k$, and scaling factor $L$
2: **Input**: Replay buffer without reward $R$, preference dataset $D$, and current policy $\pi_\phi$
3: Sample trajectories uniformly from replay buffer $T = \{\sigma_i\}_{i=0}^{2\times L\times k-1} \sim R$
4: Calculate log-likelihood $l_i = \frac{1}{H}\sum_{(s_t,a_t)\sim\sigma_i} \log \pi_\phi(a_t \mid s_t) \quad \forall \sigma_i \in T$
5: Calculate sampling weight $w_i$ using the inverse rank of log-likelihood $l_i$
6: Sample $2\times k$ trajectories and pair them with probability $P(i) = w_i^\alpha / \sum_j w_j^\alpha$
7: Add $k$ queries with corresponding human annotations to preference dataset $D$

---

---

**Algorithm 2** CER

---

1: **Require:** Hyperparameters: current threshold $\beta_i$, step size $\delta$, decay rate $\rho$, and current replay ratio $r$
2: **Input:** Maximum critic output $Q_{max}$
3: **if** $Q_{max} > \beta_i$ **then**
4:     Increase replay ratio $r+=1$
5:     Increase threshold $\beta_{i+1} = \beta_i + \delta \times \rho^i$
6:     $i+=1$
7:     Reset parameters of critic $\theta$, target critic $\bar{\theta}$, and reward estimator $\psi$
8:     $Q_{max} \leftarrow -\infty$
9: **end if**

---

**Algorithm 3** PoLiCER

---

1: **Require:** Hyperparameters: reward learning frequency $F$
2: Initialize parameters of policy $\phi$, critic $\theta$, target critic $\bar{\theta}$, and reward estimator $\psi$
3: Initialize a preference dataset $D \leftarrow \emptyset$
4: (Optional) Initialize replay buffer $R$ and policy $\pi_\phi$ with unsupervised exploration
5: $Q_{max} \leftarrow -\infty$
6: **for** each iteration **do**
7:     **if** iteration $\%F == 0$ **then**                                  $\triangleright$ feedback session
8:         Sample queries using likelihoods of current policy $\pi_\phi$      $\triangleright$ Algorithm 1
9:         Determine whether to reset or not using maximum critic output $Q_{max}$   $\triangleright$ Algorithm 2
10:        Optimize $\psi$ with Equation 2 and temporal data augmentation
11:        Relabel replay buffer $R$ using $\hat{r}_\psi$
12:     **end if**
13:     **if** feedback session over **then**             $\triangleright$ revert replay ratio after all feedback provided
14:         replay ratio $r = 1$
15:     **end if**
16:     **for** each gradient step in $r$ **do**                   $\triangleright$ agent update with current replay ratio
17:         Sample random mini-batch $B$ from $R$
18:         Optimize $\theta$, $\bar{\theta}$, and $\phi$, and log current critic outputs $Q_\theta(s_t, a_t)$ of $B$
19:         **if** $Q_\theta(s, a) > Q_{max}$ **then**        $\triangleright$ update maximum critic output for Algorithm 2
20:             $Q_{max} \leftarrow Q_\theta(s, a)$
21:         **end if**
22:     **end for**
23:     Take action $a_t \sim \pi_\phi(a_t \mid s_t)$ and collect $s_{t+1}$
24:     Store transition $\{s_t, a_t, s_{t+1}, \hat{r}_\psi(s_t, a_t)\}$ in $R$
25: **end for**

---

## C    EXPERIMENTAL DETAILS

### C.1    IMPLEMENTATION DETAILS FOR VECTOR-BASED CONTROL

We adopted SAC (Haarnoja et al., 2018) for vector-based control, with the full set of hyperparameters listed in Table 1. For the baseline PbRL method PEBBLE (Lee et al., 2021b), which uses SAC as its backbone, we employed an ensemble estimator comprising three multi-layer perceptron (MLP) models. Each MLP contained three hidden layers of 256 units with leaky rectified linear unit (ReLU) activations. The outputs of each ensemble member were bounded within the range [-1, 1] using a hyperbolic tangent activation. We trained the models using BCE loss with the Adam optimizer (Kingma & Ba, 2015), a batch size of 128, and a learning rate of 0.0003 (see Table 2). These settings closely followed those described in B-Pref benchmark (Lee et al., 2021a). For other PbRL methods built upon PEBBLE—SURF (Park et al., 2022), RUNE (Liang et al., 2022), MRN (Liu et al., 2022), and QPA (Hu et al., 2024)—we maintained the same hyperparameters as in the PEBBLE configuration. The remaining method-specific hyperparameters are provided in Tables 3 to 6. Table 7 summarizes the settings of PoLiCER, which we kept largely consistent across environ-

ments to ensure generalizability. All experiments were conducted using ten random seeds, and the environments used are illustrated in Figure 11.

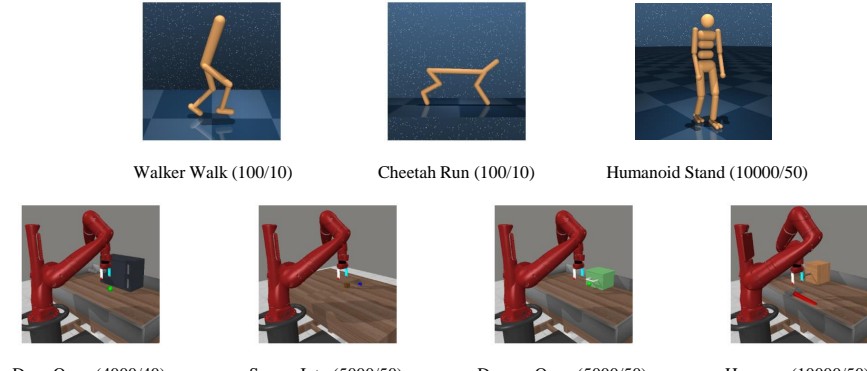

| Walker Walk (100/10) | Cheetah Run (100/10) | Humanoid Stand (10000/50) |

| Door Open (4000/40) | Sweep Into (5000/50) | Drawer Open (5000/50) | Hammer (10000/50) |

Figure 11: Visualization of test environments for vector-based control. The top row shows three locomotion tasks from DMControl: Walker Walk, Cheetah Run, and Humanoid Stand. The bottom row displays four manipulation tasks from Meta-World: Door Open, Sweep Into, Drawer Open, and Hammer.

Across all experiments, we used different query sampling strategies depending on the method. For most methods (except QPA and PoLiCER), we employed disagreement-based sampling (DS), which leverages the uncertainty of ensemble reward model to select informative queries (Christiano et al., 2017; Lee et al., 2021b). Specifically, DS operates in two steps: first, it randomly samples $L \times K$ trajectory pairs from a query buffer containing the 100 most recent episodes; then it computes the variance of preference predictions across ensemble members for each pair. High variance indicates that the reward model is uncertain about which trajectory is better, signaling regions where the model would benefit from additional training data. The top $K$ pairs with the highest variance are then selected for human annotation. Importantly, this variance measures model disagreement about the reward function, not the reliability of consistency of human annotation. Following prior studies, we set $L = 10$ for this initial sampling pool.

Table 1: Hyperparameters of SAC for vector-based control.

| Hyperparameter | Value | Hyperparameter | Value |
| --- | ---: | --- | ---: |
| Init temperature | 0.1 | Hidden dimension | 1,024 (DMControl), |
| Learning rate for agent | 0.0005 (Walker Walk, Cheetah Run), | | 256 (Meta-World) |
| | 0.0001 (Humanoid Stand), | # of layers for agent | 2 (DMControl), |
| | 0.0003 (Meta-World) | | 3 (Meta-World) |
| Target critic update frequency | 2 | Batch size for agent | 1,024 (DMControl), |
| Optimizer | Adam (Kingma & Ba, 2015) | | 512 (Meta-World) |
| Optimizer temperatures | (0.9, 0.999) | Critic EMA rate | 0.005 |
| | | Discount factor | 0.99 |

Table 2: Hyperparameters of PEBBLE for vector-based control.

| Hyperparameter | Value | Hyperparameter | Value |
| --- | ---: | --- | ---: |
| Batch size for $\psi$ | 128 | Length of segment | 50 (DMControl), |
| Total feedbacks / | 100/10 (Walker Walk, Cheetah Run), | | 25 (Meta-World) |
| # of queries per session | 4,000/40 (Door Open), | Frequency of feedback | 20,000 (Walker Walk, Cheetah Run), |
| | 5,000/50 (Drawer Open, Sweep Into), | | 5,000 (Humanoid Stand, Meta-World) |
| | 10,000/50 (Humanoid Stand, Hammer) | Pre-training steps | 9,000 |
| Learning rate for $\psi$ | 0.0003 | Data collection steps | 1,000 |
| Query buffer size | 100 | | |

For QPA, we implemented NOS as specified in their original work (Hu et al., 2024), which required reducing the query buffer size to focus on more recent experiences (see Table 6 for specific buffer size). In contrast, PoLiCER used our PLS method, which selects queries based on their relevance to

Table 3: Hyperparameters of SURF for vector-based control.

| Hyperparameter | Value | Hyperparameter | Value |
|---|---|---|---|
| Unlabeled batch ratio $\mu$ | 4 | Segment length before cropping | 60 (DMControl), |
| Loss weight for unlabeled data $\lambda$ | 1 | | 35 (Meta-World) |
| Threshold | 0.999 (Cheetah, Sweep Into), | Min/Max length of cropped segment | [45, 55] (DMControl), |
| | 0.99 (Others) | | [20, 30] (Meta-World) |

Table 4: Hyperparameters of RUNE for vector-based control.

| Hyperparameter | Value |
|---|---|
| Initial weight for intrinsic reward | 0.05 |
| Decaying rate | 0.00001 |

Table 5: Hyperparameters of MRN for vector-based control.

| Hyperparameter | Value |
|---|---|
| Meta-step frequency | 1,000 (Walker Walk) |
| | 10,000 (Door Open, Sweep Into) |
| | 5,000 (Others) |

Table 6: Hyperparameters of QPA for vector-based control.

| Hyperparameter | Value | Hyperparameter | Value |
|---|---|---|---|
| Query buffer size | 10 (DMControl), | Data augmentation ratio $\tau$ | 20 |
| | 60 (Door Open), | Hybrid experience replay sample ratio $\omega$ | 0.5 |
| | 30 (Others) | | |

Table 7: Hyperparameters of PoLiCER for vector-based control.

| Hyperparameter | Value | Hyperparameter | Value |
|---|---|---|---|
| Sampling weight temperature $\alpha$ | 1.0 | Data augmentation ratio $\tau$ | 20 |
| Initial reset threshold $\beta_0$ | 25 | Threshold growth decay rate $\rho$ | 0.9 |
| Numerator of step size $\delta$ | 30 | Maximum replay ratio $r$ | 4 |
| Denominator of step size $\delta$ | $|A|$ | | |

the current policy without restricting the buffer size. This allowed PoLiCER to maintain the default buffer size of 100 episodes while still ensuring policy-aligned queries.

For the CER component of PoLiCER, we carefully calibrated the reset thresholds based on theoretical bounds. Since the maximum possible critic output is bounded by $\left|\frac{r}{1-\gamma}\right| = 100$ with reward scale $r = 1$ and discount factor $\gamma = 0.99$, we set the threshold range to [-100, 100]. Specifically, we used an initial threshold $\beta_0 = 25$, decay rate $\rho = 0.9$, and step size $\delta = \frac{30}{|A|}$ (scaled by action dimension), ensuring that reset thresholds remained appropriate across all environments while adapting to task complexity.

## C.2 IMPLEMENTATION DETAILS FOR PIXEL-BASED CONTROL

We extended our evaluation of PoLiCER to pixel-based state inputs, as described in Section 5.4. Following the official SURF implementation for pixel-based control, we replaced the SAC backbone with DrQ-v2 (Yarats et al., 2022) and adopted the hyperparameters detailed in Table 8. For comparison, we implemented pixel-based versions of PEBBLE and SURF according to Park et al. (2022). We also incorporated QPA into our pixel-based experiments. However, since the original implementation does not support pixel-based inputs, we adapted it using hyperparameters consistent with its vector-based version, modifying only the data augmentation ratio $\tau$. Tables 9 through 12 outline the hyperparameter differences across all methods. For PoLiCER implementation built upon DrQ-v2, we applied a selective reset strategy that only affected the last three MLP layers of Q-function and reward estimator, while preserving the image encoder parameters (Nikishin et al.,

2022). This approach maintains learned visual representations across resets. All pixel-based experiments were conducted across ten random seeds, with the test environments used illustrated in Figure 12.

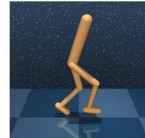 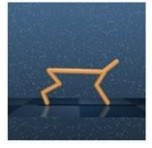 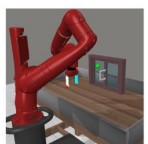

Walker Walk (200/10)   Cheetah Run (1000/50)   Window Open (800/20)

Figure 12: Visualization of test environments for pixel-based control. The left two images show locomotion tasks from DMControl: Walker Walk and Cheetah Run. The rightmost image displays a manipulation task from Meta-World: Window Open.

Table 8: Hyperparameters of DrQ-v2 for pixel-based control.

| Hyperparameter | Value | Hyperparameter | Value |
|---|---|---|---|
| Replay buffer capacity | 1,000,000 | Optimizer | Adam (Kingma & Ba, 2015) |
| Action repeat | 2 | Critic EMA rate | 0.01 |
| # of seed frames | 4,000 | Update frequency | 2 |
| Exploration steps | 2,000 | Feature dimension | 50 |
| $n$-step returns | 3 (Cheetah Run), | Hidden dimension | 1,024 |
| | 1 (Others) | Exploration std clip | 0.3 |
| Batch size | 256 (Cheetah Run), | Exploration std schedule | (1.0, 0.1, 500,000) (Cheetah Run), |
| | 512 (Others) | | (1.0, 0.1, 100,000) (Others) |

Table 9: Hyperparameters of PEBBLE for pixel-based control.

| Hyperparameter | Value | Hyperparameter | Value |
|---|---|---|---|
| Batch size for $\psi$ | 16 | Length of segment | 50 (DMControl), |
| Total feedbacks / | 200/10 (Walker Walk), | | 25 (Meta-World) |
| # of queries per session | 1,000/50 (Cheetah Run), | Frequency of feedback | 30,000 (DMControl), |
| | 800/20 (Window Open) | | 10,000 (Meta-World) |
| Learning rate for $\psi$ | 0.0003 | Query buffer size | 20 |

Table 10: Hyperparameters of SURF for pixel-based control.

| Hyperparameter | Value | Hyperparameter | Value |
|---|---|---|---|
| Segment length before cropping | 60 (Cheetah Run), | Min/Max length of cropped segment | [45, 55] (Cheetah Run), |
| | 54 (Walker Walk), | | [48, 52] (Walker Walk), |
| | 29 (Window Open), | | [23, 27] (Window Open), |
| Threshold | 0.99 | Unlabeled batch ratio $\mu$ | 5 (DMControl), |
| Loss weight for unlabeled data $\lambda$ | 1 (Cheetah Run), 0.1 (Others) | | 10 (Meta-World) |

Table 11: Hyperparameters of QPA for pixel-based control.

| Hyperparameter | Value | Hyperparameter | Value |
|---|---|---|---|
| Query buffer size | 10 | Data augmentation ratio $\tau$ | 10 |
| Hybrid experience replay sample ratio $\omega$ | 0.5 | | |

Table 12: Hyperparameters of PoLiCER for pixel-based control.

| Hyperparameter | Value | Hyperparameter | Value |
|---|---|---|---|
| Sampling weight $\alpha$ | 1.0 | Data augmentation ratio $\tau$ | 10 |
| Init threshold $\beta_0$ | 25 | Threshold growth decay rate $\rho$ | 0.9 |
| Numerator of step size $\delta$ | 30 | Maximum replay ratio $r$ | 2 (Meta-World), |
| Denominator of step size $\delta$ | $|A|$ | | 4 (DMControl) |
| Query buffer size | 30 | | |

# D ADDITIONAL EXPERIMENTAL RESULTS

## D.1 COMPARISON OF SAMPLING SCHEMES

We compared our proposed PLS with two existing sample schemes: DS and NOS. DS selects the top $K$ trajectory pairs with the highest variance among ensemble members, while NOS prioritizes queries from recent episodes to mitigate query-policy misalignment (Christiano et al., 2017; Hu et al., 2024). Details for both methods are provided in Appendix C.1.

To ensure fair comparison, we evaluated all three methods on Meta-World Sweep Into across ten runs with identical hyperparameters. We set a total feedback budget of 10,000 preferences, delivered in batches of 50 instances every 5,000 environment steps following the unsupervised pre-training phase. Performance was assessed using two metrics: learning curves (success rate) and the average log-likelihood of sampled trajectories under the current policy $\pi$ at each feedback session (Hu et al., 2024).

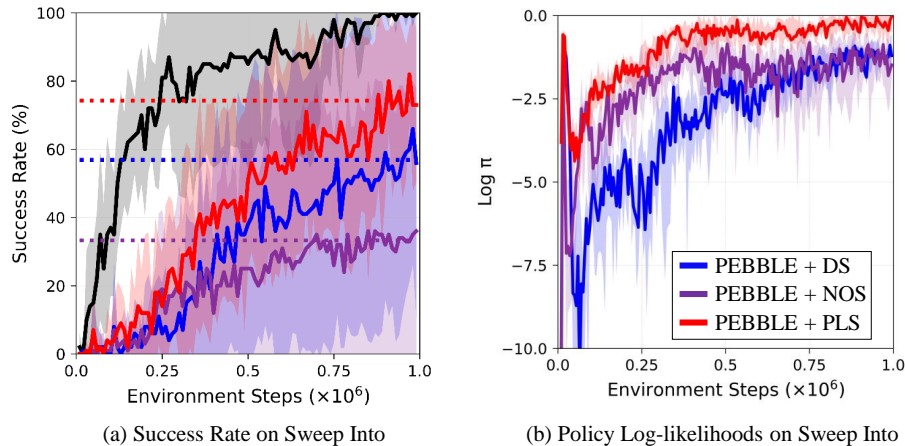

(a) Success Rate on Sweep Into      (b) Policy Log-likelihoods on Sweep Into

Figure 13: Comparison of three different query sampling strategies on Meta-World Sweep Into (10,000/50) with ten random seeds. The feedback session configuration is consistent with Figure 1. (a) Success rate learning curves over ten evaluation episodes, showing the mean and standard deviation across runs. (b) Average log-likelihoods of sampled trajectories under the current policy $\pi$ at each feedback session. Median and IQR are reported to mitigate the influence of outliers.

As shown in Figure 13, PLS significantly improved the success rate by approximately 20% compared to the competing methods and consistently achieved the highest policy log-likelihoods. While NOS demonstrated effectiveness early in training, its benefits diminished over time, with the gap in policy likelihood between NOS and DS gradually narrowing. This suggests that recency becomes a less reliable proxy for policy relevance as task complexity increases. Importantly, PLS introduces no noticeable computational overhead. On an RTX 4090 GPU with an i9-13900KF CPU, PLS completed one million training steps in approximately 1.5 hours, comparable to the training time required by NOS.

Table 13: Actor entropy statistics for three sampling strategies on Meta-World Sweep Into (10,000/50) with ten random seeds.

|      | DS     | NOS    | PLS    |
| ---- | ------ | ------ | ------ |
| Mean | -3.688 | -3.759 | -3.758 |
| Std  | 1.138  | 0.948  | 0.945  |
| Max  | 2.236  | 2.209  | 2.267  |
| Min  | -4.271 | -4.158 | -4.135 |

One might consider that PLS could implicitly bias sampling toward low-entropy policies due to its localized guidance. However, policy entropy is controlled independently by the backbone RL

algorithm (i.e., SAC uses an automatic entropy adjustment, and DrQ-v2 relies on scheduled exploration noise). Therefore, the sampling strategies do not directly influence entropy dynamics. To empirically verify this, we compared policy entropy statistics across three sampling strategies used in our experiments. As shown in Table 13, entropy remains nearly identical regardless of sampling methods, indicating that PLS does not induce low-entropy behavior.

## D.2 FURTHER ANALYSIS OF REWARD OVERESTIMATION

We further analyzed the detrimental effect of reward overestimation caused by primacy bias, as discussed in Section 4.2. While Figure 2 demonstrates how overestimation persists in PEBBLE through learning curves and overestimation rates, Figure 14 presents a more detailed visualization by comparing estimated and true rewards over 1,000 collected episodes in Walker Walk. We carefully sampled these episodes to ensure a uniform distribution of true rewards ranging from 0 to 1,000 (see Table 14 for detailed statistics).

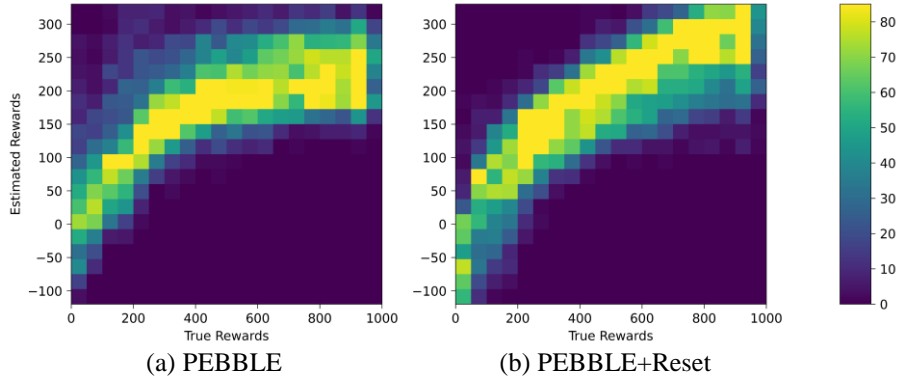

(a) PEBBLE      (b) PEBBLE+Reset

Figure 14: Comparison between estimated rewards and true rewards for 1,000 collected episodes in Walker Walk, uniformly sampled across the full reward range. (a) PEBBLE exhibits strong reward overestimation, particularly in low-reward regions. (b) PEBBLE+Reset alleviates this problem, producing more accurate reward estimates across the full reward range.

The heatmap reveals that PEBBLE exhibited severe reward overestimation, particularly for low-reward trajectories, where early feedback disproportionately inflated reward estimates. In contrast, PEBBLE+Reset produced a more accurate and consistent relationship between estimated and true rewards. Additionally, PEBBLE struggled to distinguish between high- and mid-reward trajectories because of compressed estimated values in those regions, limiting the reward estimator's ability to capture performance differences needed for effective policy optimization.

Table 14: Statistics for true episode rewards collected in Walker Walk.

| Statistic | Value |
|---|---|
| Count | 1,000 |
| Average | 500.168 |
| Standard deviation | 281.716 |
| Minimum | 19.826 |
| 25th percentile (Q1) | 257.404 |
| 50th percentile (Q2) | 500.595 |
| 75th percentile (Q3) | 744.916 |
| Maximum | 989.187 |

As described in Section 1 and 4.2, resetting the reward estimator at every feedback session helped mitigate reward overestimation. However, this approach alone proved insufficient because the Q-function retained biased estimates from previously overestimated rewards. Our CER method addressed this by resetting both the reward estimator and Q-function when current critic outputs exceeded a dynamically increasing threshold.

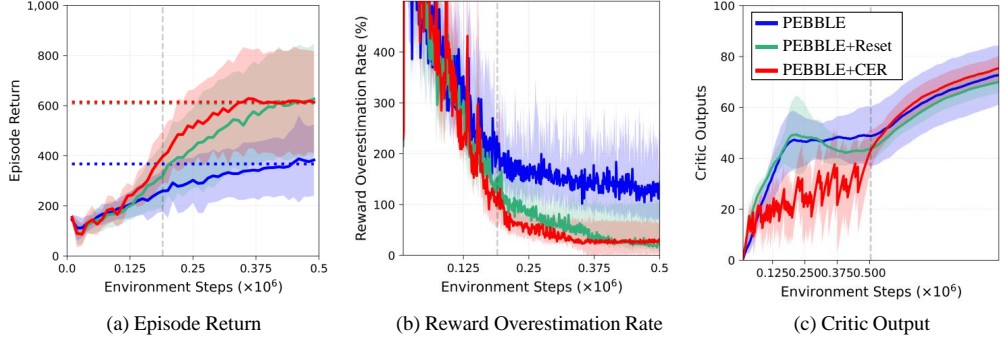

(a) Episode Return     (b) Reward Overestimation Rate     (c) Critic Output

Figure 15: Performance comparison between PEBBLE, PEBBLE + Reset, PEBBLE + CER on Walker Walk. (a) Learning curves showing the average episode return, with the mean and standard deviation across ten runs. (b) Reward overestimation rate during training, displaying median and IQR. (c) Critic output magnitudes representing Q-value estimates, reported as mean and standard deviation. CER achieved both faster return gain and more effective mitigation of overestimation.

Figure 15 demonstrates the effectiveness of CER. Figures 15 (a) and (b) reveal that agents trained with CER achieved higher returns and lower reward overestimation rates, respectively, compared to using reward estimator reset alone. Importantly, Figure 15 (c) shows that CER had a significant impact on reducing Q-function overestimation, resulting in more accurate value estimates throughout training.

## D.3 ADDITIONAL RESULTS FOR VECTOR-BASED CONTROL

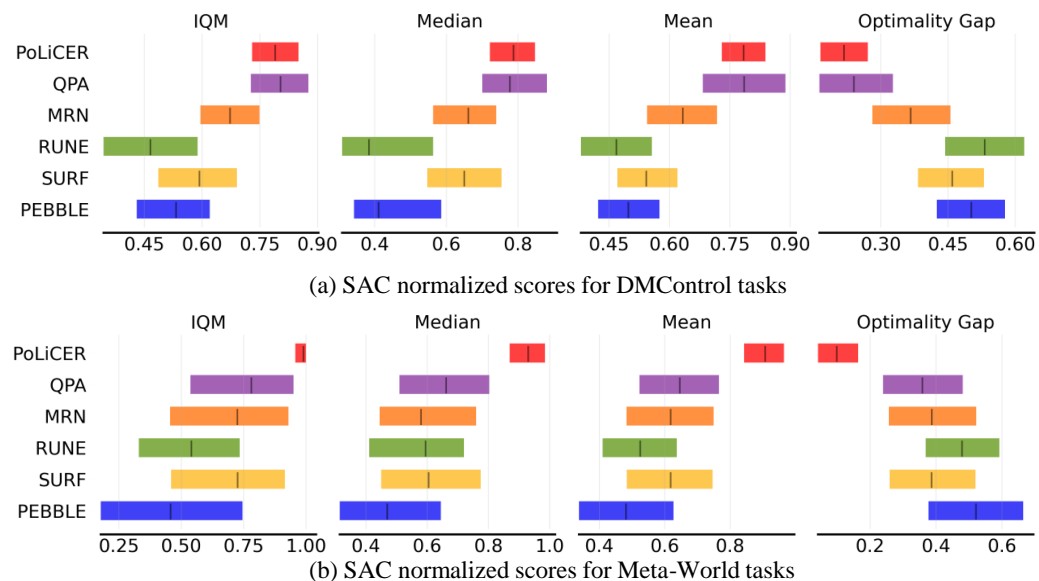

(a) SAC normalized scores for DMControl tasks

(b) SAC normalized scores for Meta-World tasks

Figure 16: Aggregate performance metrics for all methods across ten independent runs. (a) Results for three DMControl tasks. (b) Results for four Meta-World tasks. Each panel shows SAC-normalized IQM, Median, Mean and OG with 95% stratified bootstrap confidence intervals. Lower OG values indicate performance closer to SAC with ground-truth rewards.

To ensure robust and reliable performance evaluation, we used *rliable* library (Agarwal et al., 2021) to compute several normalized metrics relative to SAC: interquartile mean (IQM), mean, median, and optimality gap (OG). IQM represents the average performance within the interquartile range (middle 50% of runs), effectively reducing influence of outliers. OG quantifies the deviation from

optimal performance (SAC in our study), with lower values indicating closer alignment to optimal results. We reported all metrics with 95% stratified bootstrap confidence intervals. Figures 16 (a) and (b) present the aggregated results across three DMControl tasks and four Meta-World tasks, respectively. Across both benchmarks, PoLiCER consistently outperformed other PbRL methods, achieving the highest IQM and lowest OG in Meta-World. Although PoLiCER and QPA yielded similar average scores in DMControl, PoLiCER showed significantly narrower 95% CI bands, demonstrating greater stability across tasks. The lower OG values further confirmed that PoLiCER remained closer to SAC performance, highlighting its efficiency under limited feedback conditions.

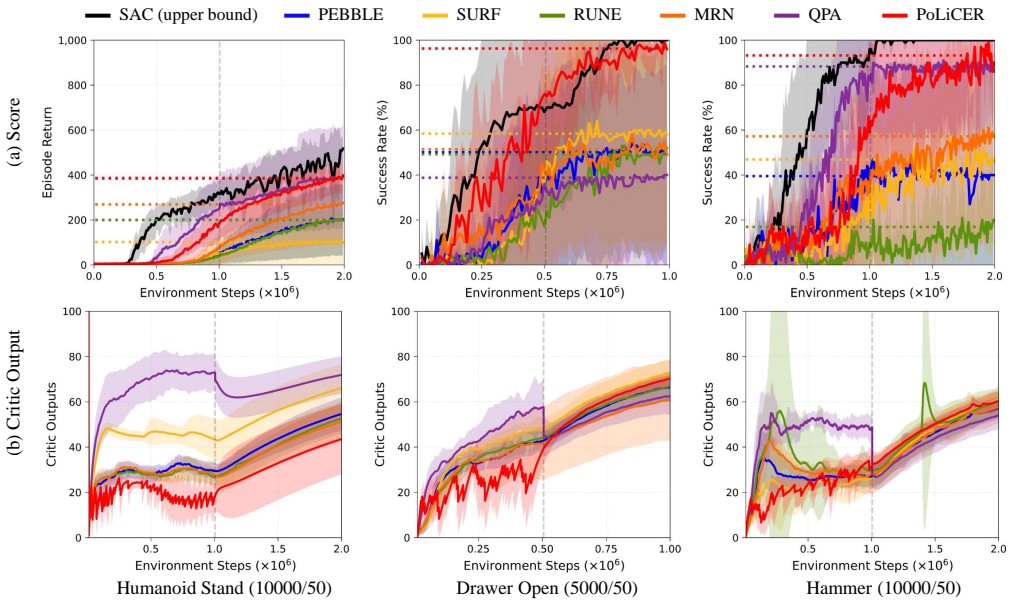

Figure 17: Results across ten runs on Humanoid Stand, Drawer Open, and Hammer. (a) Learning curves showing performance progression with horizontal dotted lines indicating final performance. (b) Critic outputs during training. In both rows, solid lines represent means, shaded areas show standard deviation.

Building on these aggregate results, we next visualized the learning curves and corresponding critic output plots across all methods on three challenging tasks—Humanoid Stand, Drawer Open, and Sweep Into (Figure 17). Solid lines represent mean values, shade regions indicate standard deviation, and vertical dashed gray lines mark the last feedback step. Dotted lines in the learning curves denote the final performance, calculated by averaging the last ten evaluation scores. The critic output plots show that most baselines suffered from overestimation during the feedback phase, with value estimates inflated beyond actual performance (Van Hasselt et al., 2016). Notably, QPA exhibited the highest critic outputs, driven by its use of recent—and thus policy relevant—data for query sampling and Q-function updates, reinforcing localized guidance (Hu et al., 2024). In contrast, despite also using policy-aligned sampling, PoLiCER maintained much more stable critic values. This highlights the effectiveness of CER in mitigating overestimation and stabilizing training (Nikishin et al., 2022). This enabled PoLiCER to combine feedback efficiency with robustness, avoiding runaway value estimates while preserving policy relevance.

## D.4 DETRIMENTAL EFFECT OF REWARD MISIDENTIFICATION

Figure 18 illustrates problematic cases in the Drawer Open environment where the true reward signal significantly diverged from actual task success, creating misleading preference feedback. Figure 18 (a) shows a successful task completion—the robot eventually opened the drawer completely—but received unexpectedly low true rewards because of time spent exploring before achieving the goal. Conversely, Figure 18 (b) depicts a clear failure case where the robot merely grasped the drawer handle without fully opening it, yet paradoxically received high true rewards.

This reward-task misalignment led synthetic annotators to generate inconsistent preference labels that poorly reflected actual task completion. As documented by Lee et al. (2021a), such noisy preferences can severely impact learning by misguiding both policy updates and query selection processes. This problem created a detrimental feedback loop where the query sampler reinforced suboptimal behaviors based on flawed reward signals. This phenomenon helps explain high performance variance and inconsistent learning outcomes observed across most baseline methods.

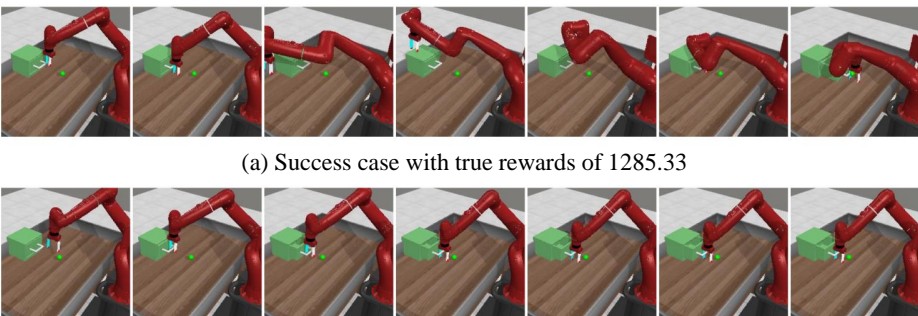

(a) Success case with true rewards of 1285.33

(b) Failure case with true rewards of 3013.11

Figure 18: Visualization of reward-task misalignment in Drawer Open. (a) Successful drawer opening that received unexpectedly low rewards. (b) Failed attempt that received disproportionately high rewards. This misalignment generated noisy synthetic preferences that hindered consistent policy learning.

## D.5 COMPUTATIONAL OVERHEAD ANALYSIS FOR PIXEL-BASED CONTROL

Table 15: Wall-clock training time (hours) comparison for pixel-based control tasks across PbRL methods.

| Method | Walker Walk (1M frames) (200/10) | Cheetah Run (1M frames) (1000/50) | Window Open (0.8M frames) (800/20) |
|---|---|---|---|
| DrQ-v2 (upper bound) | 3.47 | 2.53 | 7.24 |
| PEBBLE | 4.04 | 3.21 | 8.77 |
| SURF | 4.27 | 4.36 | 11.45 |
| QPA | 4.26 | 3.28 | 8.22 |
| PoLiCER | 5.88 | 4.68 | 9.35 |

We analyzed the computational overhead of PoLiCER for the pixel-based control experiments described in Section 5.4. Wall-clock training time was measured across methods on three representative tasks using an RTX 4090 GPU with an i9-13900KF CPU. RL with pixel-based states is inherently computationally expensive due to environment rendering. DrQ-v2 employs action repeat of 2, executing each action for 2 environment steps with a single rendering. In our setup, this results in approximately 11–12 seconds per 1,000 environment frames in DMControl and 14–15 seconds per 500 frames in Meta-World. This rendering cost dominates training time, with DrQ-v2 (ground-truth rewards) requiring 2.5–3.5 hours for 1M frames in DMControl and 7.2 hours for 0.8M frames in Meta-World Window Open. As shown in Table 15, all PbRL methods incur additional overhead from reward model updates, with SURF requiring extra computation for semi-supervised learning and PoLiCER introducing overhead through CER-triggered increased replay ratio after resets. PoLiCER's overhead primarily stems from CER rather than PLS. PLS adds only $2 \times L \times K$ policy forward passes per feedback session (Appendix D.1), substantially less than DS. CER dynamically increases the replay ratio after resets during the feedback collection phase, returning to 1 once all feedback is collected (Algorithm 3, lines 13–15). Therefore, The overhead magnitude varies across feedback schedule and reset frequency. While PoLiCER incurred approximately 8–9% additional training time, it achieved approximately $80\%$ success rate, doubling the performance of PEBBLE

on Window Open. This overhead remains modest relative to the inherent rendering costs and reward learning that dominate pixel-based PbRL.

### D.6 HUMAN-IN-THE-LOOP EXPERIMENT PROCEDURE

We recruited volunteers consisting of undergraduate and graduate students not majoring in RL or robotics, ensuring preference judgments reflect intuitive human understanding of natural motion without domain expertise. Each participant annotated trajectories for a single run of both PEBBLE and PoLiCER to ensure unbiased comparison (Muslimani & Taylor, 2025). At each feedback session, participants were presented with pairs of trajectory videos displayed side-by-side for direct comparison. Each video clip spanned a segment length of 50, corresponding to 2.5 seconds at 20 frames per second (fps). We used a lateral camera view at $96 \times 192$ resolution for Walker Walk, while employing a dual-view setup combining corner and top-down perspectives vertically into a single $288 \times 288$ resolution for Window Open, as the side view alone proved insufficient for judging spatial relationships between the robot arm and window handle (refer to Figure 19). Participants could replay videos repeatedly before making a decision. After 10,000 steps of unsupervised pre-training (Lee et al., 2021b), we collected preference labels at regular intervals throughout training. For Walker Walk, we gathered 10 queries every 20,000 steps (totaling 100 labels), while Window Open required 15 queries every 10,000 steps (totaling 240 labels). These intervals correspond to approximately 10 minutes of wall-clock time in both environments. Completing all feedback sessions required approximately 1.5–1.8 hours for Walker Walk and 2.6–3.0 hours for Window Open.

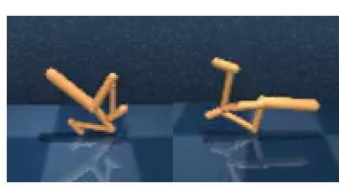 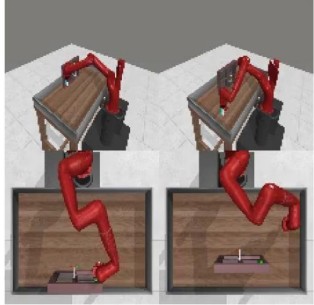

Walker Walk                                    Window Open

Figure 19: Snapshot examples of query pairs to human annotators in Walker Walk (left) and Window Open (right).

To ensure consistent and meaningful feedback, we provided clear task-specific criteria to annotators for each environment. Participants were instructed to view both trajectory clips simultaneously and select the relatively better trajectory. We informed them that their feedback was actively shaping the agent's learning and that both clips might show suboptimal behavior early in training. While participants were encouraged to choose the relatively better one even in such cases, an equal preference option was available for genuinely ambiguous cases where trajectories appeared indistinguishable in quality. Annotators were instructed to evaluate trajectories in the following priority order:

**Walker Walk**

- If fallen, prefer the trajectory that recovers to standing position more rapidly
- If moving forward, prefer coordinated alternating leg motion rather than limping or relying on one leg
- When standing or moving forward, prefer maintenance of upright torso posture.

**Window Open**

- At the initial stage with the window closed, prefer trajectories where the end-effector moves closer to the handle

- If the handle is contacted, prefer leftward sliding motion that fully opens the window (from corner view)

- Throughout manipulation, prefer smooth and direct movements without excessive jitter or unnecessary detours

While we pursued a fair and controlled HiL experiment by providing concrete evaluation criteria, some exceptional cases not specified led participants to express varying preferences for similar query pairs, resulting in mixed or personalized preferences. We present two representative examples in Window Open below to identify critical considerations for improving guideline design and annotation consistency in real-human experiments. Figure 20 (a) illustrates state-action credit assignment ambiguity, where one trajectory showed active sliding motion with less opening currently while the other achieved greater opening but remained stationary: annotators disagreed on whether to prioritize current action or state. Figure 20 (b) shows heterogeneous stage comparison difficulty, where each trajectory performed well at different task stages (approaching vs sliding), making direct comparison ambiguous without cross-stage evaluation priorities. These cases highlight the need for more detailed guidelines that explicitly specify stage-specific priorities and clarify state-action trade-offs in evaluation criteria. Even with such mixed preferences across annotators, PoLiCER demonstrated more consistent performance than PEBBLE by mitigating primacy bias through CER's periodic resets, preventing the reward model from overfitting to individual annotator preferences.

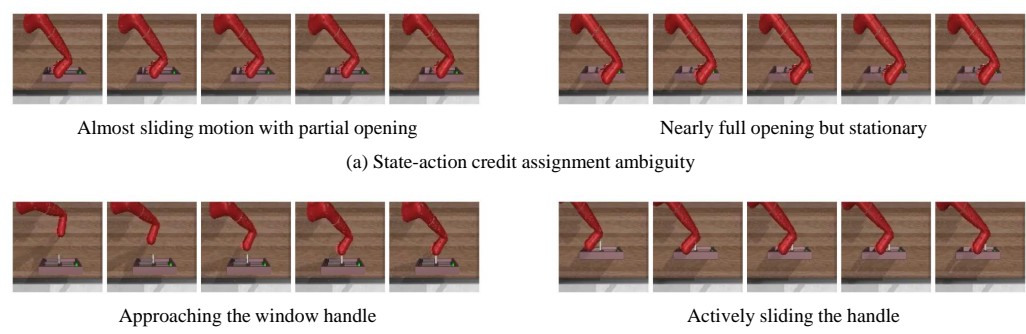

Almost sliding motion with partial opening   Nearly full opening but stationary

(a) State-action credit assignment ambiguity

Approaching the window handle   Actively sliding the handle

(b) Heterogeneous stage comparison difficulty

Figure 20: Representative examples of mixed or personalized preference cases that led to inter-subject annotation disagreement: (a) State-action credit assignment ambiguity and (b) Heterogeneous stage comparison difficulty.

### D.7 EFFECT OF HYPERPARAMETERS

We investigated PoLiCER's sensitivity to two key hyperparameters in DMControl Walker Walk; the temperature parameter $\alpha$ for PLS and the initial reset threshold $\beta_0$ for CER. Figure 21 presents the final performance achieved using 100 feedback instances across ten runs for various hyperparameter settings.

For the sampling temperature $\alpha \in \{0.0, 0.5, 1.0, 1.5, 2.0\}$, our results showed that $\alpha = 1.0$ yielded the best overall performance. This moderate $\alpha$ value effectively balanced prioritization of policy-aligned queries with sufficient exploration diversity. Very high $alpha$ values (1.5 and 2.0) provided only marginal improvements, likely because of overly concentrated sampling that limited exploration. Although not shown in the figure, we observed that higher $\alpha$ accelerated early-stage learning, but this advantage diminished over time compared to uniform sampling ($\alpha = 0.0$).

For the initial reset threshold $\beta_0$, we found a clear trade-off. Lower $\beta_0$ triggered more frequent resets, which reduced long-term bias accumulation but introduced short-term instability in TD learning (Nikishin et al., 2023). Higher $\beta_0$ reduced computational costs by decreasing reset frequency but allowed primacy bias from early, potentially low-quality feedback to persist longer. Despite these competing factors, PoLiCER maintained consistent performance across a broader range of $\beta_0$ values, demonstrating robust performance without requiring precise hyperparameter tuning.

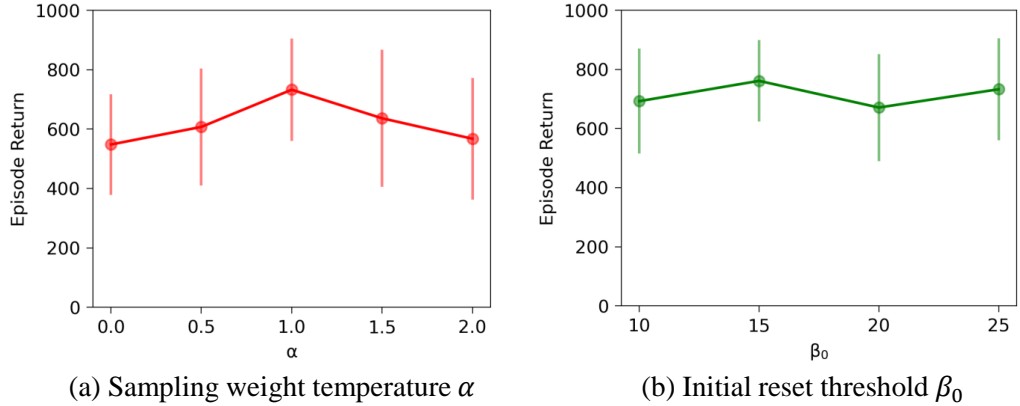

(a) Sampling weight temperature $\alpha$       (b) Initial reset threshold $\beta_0$

Figure 21: Final performance on Walker Walk for PoLiCER using 100 feedback instances across ten runs: (a) Effect of sampling weight temperature $\alpha$ for PLS and (b) Effect of initial reset threshold $\beta_0$ for CER. Error bars indicate standard deviation across runs.

## E   LARGE LANGUAGE MODEL USAGE

Large Language Model (Claude, developed by Anthropic) was used to aid and polish the writing of this manuscript. Specifically, the LLM was used for:

- Grammar checking and correction
- Improving sentence clarity and readability
- Suggesting alternative phrasings for better expression

The LLM was used solely as a writing assistance tool and did not contribute to research ideation, methodology development, experimental design, data analysis, or any core technical contributions. All research ideas, algorithmic innovations, experimental results, and scientific conclusions are entirely the work of the human authors.

