# OpenReview forum: "Policy Likelihood-based Query Sampling and Critic-Exploited Reset for Efficient Preference-based Reinforcement Learning"
_ICLR.cc/2026/Conference — ICLR 2026 Poster_

### Official Review · Reviewer_SQmv · 2025-10-25

**Soundness:** 3
**Presentation:** 3
**Contribution:** 3
**Rating:** 8
**Confidence:** 4

**Summary:**

This paper presents a new method called PoLiCER to improve query efficiency in previous PbRL work. It proposes two improvements: policy likelihood sampling (PLS) aims to make the reward model focus on the trajectories that are likely under the current policy; critic exploited reset forces the reward estimator to reset occasionally to reduce overestimation bias that emerges from repeated feedback over the same region of behaviour. Experiments involved in locomotion and manipulation show that PoLiCER outperforms prior PbRL algorithms.

**Strengths:**

1. Preference-based RL is a growing area with practical relevance. This work provides a new perspective to improve the algorithms, promoting the development of the field.

2. The motivation is clear, and the authors use experiments to demonstrate that these issues are indeed important, which enhances the readability and coherence of the paper.

3. The two mechanisms the authors propose are beneficial to each other: reset the critic and reward model to make them more plastic to adapt to preference data that is more consistent with the current strategy.

4. The authors provide empirical evaluation across multiple tasks (locomotion + robot manipulation), which helps test generality.

5. The code is included in the Supplementary Material, demonstrating the reproducibility of this work.

**Weaknesses:**

1. PLS seems to lack some theoretical insight to support. It is not clear why directly maximizing the policy likelihood can find more informative trajectories.

2. Human-in-the-loop experiments are relatively simple.

3. In the main experiments, using TA as another trick to the algorithm may cause SURF to completely become an ablation of this method, and the additional setting of the ratio may lead to unfairness in the experiment.

**Questions:**

1. What are the details in human-in-the-loop experiments? How did the volunteers provide feedback? I hope the authors can provide some examples to prove that their experiments are fair, i.e., this feedback may show personalized preferences or mixed preferences, and PoLiCER can handle them. Meanwhile, explain what factor will affect the process.

2. In the main experiments, why is QPA better than PoLiCER in Hammer? PLS improves QPA's sampling method, so PoLiCER should definitely be better than QPA in all experiments, but the results are not.

3. What is the time consumption for pixel-based tasks? Since these tasks generally take more time, it is important to report your time consumption and compare it with previous methods.

---

> ### Author Response · Authors · 2025-11-22
> **Response to Reviewer SQmv (Part 1/3)**
>
> We sincerely appreciate the reviewer’s constructive feedback. Your questions have significantly enhanced the experimental rigor and transparency of our work, particularly regarding human-in-the-loop procedures and computational analysis. We address each comment in detail below and believe the revisions fully resolve the raised concerns. All changes in the revised manuscript are highlighted in blue to support easy verification.
>
> ---
>
> >**W1. PLS seems to lack some theoretical insight to support. It is not clear why directly maximizing the policy likelihood can find more informative trajectories.**
>
> #### We are happy to clarify this point. In PbRL, queries are informative when they help the reward model provide accurate feedback for current policy learning. The key challenge is distributional mismatch: if the reward model is trained on queries that differ from the distribution the policy visits, it produces unreliable predictions when evaluating trajectories the policy generates. PLS directly addresses this by prioritizing queries the current policy frequently generates, ensuring the reward model learns on data aligned with the policy's distribution. Appendix D.1 empirically validates that PLS achieves superior task performance compared to other sampling strategies, indicating that it identifies more informative queries within limited feedback budgets. ####
>
> ---
>
> >**W2. Human-in-the-loop experiments are relatively simple.**
>
> #### We acknowledge that human-in-the-loop (HiL) experiments are relatively simple and broader validation is essential. To strengthen the empirical evidence, we have added HiL experiments on Window Open and relocated the qualitative comparison for Walker Walk from Appendix into the main paper (please refer to Section 5.5 and response to Q1). While Window Open experiments are currently based on five runs due to the rebuttal timeline, we plan to extend them to ten runs in the final version. ####
>
> ---
>
> >**W3. In the main experiments, using TA as another trick to the algorithm may cause SURF to completely become an ablation of this method...**
>
> #### Thank you for this important question. We clarify that SURF and PoLiCER represent fundamentally different approaches: SURF employs semi-supervised learning with both labeled and unlabeled queries, while PoLiCER uses supervised learning with labeled queries only. Therefore, SURF is not an ablation of our method but a distinct experimental paradigm. Regarding TA configurations, SURF originally introduced TA for semi-supervised learning with unlabeled queries (ratio=1), which we faithfully replicated. QPA subsequently adapted TA to apply multiple augmentations to each labeled query for improved data efficiency (ratio=20), and we adopted this formulation as QPA represents the most relevant baseline. Nevertheless, we acknowledge the reviewer's concern regarding TA's contribution to PoLiCER's performance. To address this, Section 5.3 provides an ablation demonstrating that performance gains primarily stem from our proposed PLS and CER mechanisms rather than TA itself, as PoLiCER without TA still outperforms SURF by a meaningful margin (SURF in Figure 5 vs. PoLiCER w/o TA in Figure 6). ####

---

> ### Author Response · Authors · 2025-11-22
> **Response to Reviewer SQmv (Part 2/3)**
>
> >**Q1. What are the details in human-in-the-loop experiments? How did the volunteers provide feedback?...**
>
> #### Thank you for this important question. We have substantially expanded the HiL experimental procedure in revised Appendix D.6. We briefly outline the essential aspects: All participants were non-expert students who evaluated both PEBBLE and PoLiCER through matched annotator design, viewing side-by-side video comparisons (2.5s clips). For Window Open, we employed a dual-view setup to facilitate spatial judgment (Figure 19). We provided clear task-specific evaluation guidelines with situation-dependent criteria as below: ####
>
> #### **Walker Walk** ####
> - #### If fallen, prefer the trajectory that recovers to standing position more rapidly ####
> - #### If moving forward, prefer coordinated alternating leg motion rather than limping or relying on one leg ####
> - #### When standing or moving forward, prefer maintenance of upright torso posture ####
>
> #### **Window Open** ####
> - #### At the initial stage with the window closed, prefer trajectories where the end-effector moves closer to the handle ####
> - #### If the handle is contacted, prefer leftward sliding motion that fully opens the window (from corner view) ####
> - #### Throughout manipulation, prefer smooth and direct movements without excessive jitter or unnecessary detours ####
>
> #### As the reviewer noted, exceptional cases beyond our specified criteria led to mixed or personalized preferences across annotators. Figure 20 in Appendix D.6 illustrates two representative cases on Window Open: (1) state-action credit assignment ambiguity, where annotators disagreed on prioritizing current actions versus achieved states, and (2) heterogeneous stage comparison difficulty between different task phases. These cases highlight the need for more comprehensive guidelines that explicitly specify stage-specific priorities (i.e., approaching vs. manipulating) and clarify state-action trade-offs (i.e., achieved progress vs. current behavior) to minimize subjective judgment for more rigorous and reproducible experiments. Nevertheless, PoLiCER demonstrated more consistent performance than PEBBLE despite these challenges. CER's periodic resets prevent overfitting to early preferences and break the primacy bias cycle that would otherwise amplify individual annotation biases, enabling stable learning across diverse annotators. ####
>
> ---
>
> >**Q2. In the main experiments, why is QPA better than PoLiCER in Hammer?...**
>
> #### Thank you for this important question. In Hammer, QPA shows faster initial convergence but plateaus around 90% success rate due to a complete failure in one seed (zero success throughout training). In contrast, PoLiCER achieves higher final performance with non-zero success across all runs. Over the final 100 episodes (last 10 evaluation points × 10 seeds), PoLiCER reaches 93.2 ± 16.54 compared to QPA's 88.3 ± 27.35. The significantly high variance of QPA stems from the zero-success seed, reflecting the same failure mechanism as in Drawer Open: early success-misaligned feedback causes reward misidentification and localized training, leading to complete failure (analyzed in Section 5.2-3). PoLiCER mitigates this through CER, which detects overestimation and restores plasticity. ####

---

> ### Author Response · Authors · 2025-11-22
> **Response to Reviewer SQmv (Part 3/3)**
>
> ---
> >**Q3. What is the time consumption for pixel-based tasks?...**
>
> #### Thank you for this important question. We have added computational analysis to Section 5.4 and Appendix D.5. First, we not that pixel-based RL is inherently computationally expensive due to rendering costs. On our RTX 4090 GPU with i9-13900KF CPU, DMControl requires approximately 11-12 seconds per 1,000 frames, and Meta-World requires 14-15 seconds per 500 frames even without preference learning. This rendering cost dominates wall-clock time: DrQ-v2 takes 2.5-3.5 hours for 1M frames in DMControl and 7.2 hours for 0.8 frames in Meta-World. ####
>
>
> #### As shown in the Table 15, PoLiCER’s training time is comparable to other PbRL methods. All methods incur overhead from reward model updates, with SURF requiring extra computation for semi-supervised learning. PoLiCER’s overhead primarily stems from CER’s increased replay ratio after resets. PLS adds negligible overhead (Appendix D.1). While PoLiCER takes moderately longer than baselines, this cost is offset by substantial performance improvements in challenging pixel-based environments. ####
>
>
> #### **Table 15.** Wall-clock training time (hours) comparison for pixel-based control tasks across PbRL methods. ####
>
> | **Method** | **Walker Walk** | **Cheetah Run** | **Window Open** |
> |-----------|----------------|----------------|----------------|
> |  | (1M frames, 200/10) | (1M frames, 1000/50) | (0.8M frames, 800/20) |
> | DrQ-v2 (upper bound) | 3.47 | 2.53 | 7.24 |
> | PEBBLE | 4.04 | 3.21 | 8.77 |
> | SURF | 4.27 | 4.36 | 11.45 |
> | QPA | 4.26 | 3.28 | 8.22 |
> | PoLiCER| 5.88 | 4.68 | 9.35|

---

> ### Comment · Reviewer_SQmv · 2025-11-24
>
> Thank you for the detailed responses; most of my concerns have been resolved. I would like to follow up on a few of the weaknesses and your replies:
>
> w1: I would like to see a more theoretical response.
>
> w3: A small correction: QPA is indeed one of the baselines for your method. By comparing the experiments in Sections 5.3 and 5.2, I also observe a clear improvement, and therefore I am convinced of the effectiveness of your approach.
>
> I decide to keep my score.

---

> ### Author Response · Authors · 2025-11-27
> **Thank you for the constructive feedback!**
>
> We are pleased to hear that most of your concerns have been resolved. We are currently finalizing our additional human-in-the-loop experiments as described in response to Weakness 2, and will update the manuscript with comprehensive results once completed. We sincerely appreciate your thoughtful and constructive feedback, which has significantly improved this paper.
>
> ---
> **Updated:**
>
> We have finalized the human-in-the-loop experiments (10 runs) on Window Open and updated Section 5.5 accordingly.

---

### Official Review · Reviewer_vBkx · 2025-10-30

**Soundness:** 4
**Presentation:** 3
**Contribution:** 3
**Rating:** 6
**Confidence:** 4

**Summary:**

PoLiCER addresses two central challenges in preference-based RL: query-policy misalignment and reward overestimation caused by primacy bias. It proposes Policy Likelihood-based Sampling (PLS) to select queries aligned with the current policy and Critic-Exploited Reset (CER) to dynamically reset the reward model and critic when overestimation is detected.

**Strengths:**

The paper focuses on (1) query–policy misalignment in query sampling and (2) reward overestimation caused by primacy bias in reward learning, which are key issues of high importance in PBRL.

The design of the approach is reasonable, including the likelihood computation in PLS and the CRITIC-EXPLOITED RESET mechanism; it is simple and low-cost.

It shows clear performance advantages on both vector-observation and pixel-observation tasks, and provides human-in-the-loop validation along with fairly comprehensive ablations.

**Weaknesses:**

PLS is sensitive to policy entropy/scale and may prefer low-entropy trajectories, reducing diversity.

The discussion of related work could be more complete. PPE~[1] advocates proximal policy exploration to expand the coverage of the preference buffer and improve reward model quality; like this paper’s PLS, it aims to make labeled data/queries closer to the current policy, but PPE emphasizes active generation/exploration, whereas PLS passively selects from the replay buffer and ranks by likelihood. The difference and advantage of PLS over PPE should be discussed in the manuscript.

Minor issues such as typos: “primary bias” vs. “primacy bias”; a unified wording is recommended.


[1] Zhu, Y., ... . (2024). Optimizing reward models with proximal policy exploration in preference-based reinforcement learning. In NeurIPS 2024 Workshop on Behavioral Machine Learning.

**Questions:**

Does the PLS likelihood score bias toward low-entropy policies? It is recommended to add sensitivity experiments on policy entropy/temperature.

Is a diversity constraint necessary?

---

> ### Author Response · Authors · 2025-11-22
> **Response to Reviewer vBkx (Part 1/2)**
>
> Thank you for your insightful questions. Your comments helped us clarify the distinction between policy relevance and diversity, improving connections to related works such as PPE, and identifying promising future directions for expanding our method. We have carefully considered each of your points, and our detailed responses to each question and suggestions are provided below. All revisions in the manuscript are highlighted in blue for easy identification.
>
> ---
>
> > **W1. PLS is sensitive to policy entropy/scale and may prefer low-entropy trajectories, reducing diversity.**
>
> #### We agree that PLS may reduce query diversity by concentrating feedback on trajectories most aligned with the current policy’s behaviors. As shown in our ablation study (Section 5.3), using PLS alone can lead to overly concentrated queries, resulting in localized guidance and critic overestimation. This is precisely why we propose CER as a complementary component. CER mitigates these downsides by detecting overestimation and resetting both the reward estimator and Q-function when necessary, enabling the policy to explore broader regions beyond initially concentrated areas. This ensures that while PLS maintains policy relevance, the overall learning process preserves sufficient diversity to avoid suboptimal convergence. ####
> ---
>
> > **W2. The discussion of related work could be more complete. PPE [1] advocates proximal policy exploration to expand the coverage of the preference buffer and improve reward model quality...**
>
> #### Thank you for insightful suggestion. We have included a discussion of PPE [1] in the Section 2.1. PPE and PLS address query-policy misalignment through different mechanisms: PPE tackles it indirectly through exploration by adjusting the behavior policy to expand data coverage near the current policy, while PLS directly addresses it by selecting queries based on policy likelihoods from the replay buffer. PPE determines what transition data is collected, whereas PLS determines which queries to annotate from collected data. These approaches complementary: PPE enhances the replay buffer's coverage through exploration, while PLS identifies the most informative queries from this enriched dataset for annotation. We believe that combining them represents a promising direction for further improving feedback efficiency. ####
>
> ---
>
> > **W3. Minor issues such as typos: “primary bias” vs. “primacy bias”; a unified wording is recommended.**
>
> #### Thank you for your careful reading and pointing out the inconsistency. We have corrected the terminology and use “primacy bias” consistently throughout the paper to maintain clarity. ####
> ---
>
> #### **Reference** ####
> #### [1] Zhu, Y., Liu, J., Yuan, Y., Wei, W., Ge, Z., Fang, Z., ... & An, B. Optimizing Reward Models with Proximal Policy Exploration in Preference-Based Reinforcement Learning. In NeurIPS 2024 Workshop on Behavioral Machine Learning. ####

---

> ### Author Response · Authors · 2025-11-22
> **Response to Reviewer vBkx (Part 2/2)**
>
> >**Q1. Does the PLS likelihood score bias toward low-entropy policies?...**
>
> #### Thank you for raising this point. We believe this concern may stem from our discussion of "localized guidance" (Section 5.3), which relates to query diversity rather than policy entropy. We clarify that PLS does not bias toward low-entropy policies, as policy entropy is controlled independently by backbone RL algorithm rather than the query selection. In SAC (vector-based tasks), entropy is automatically maintained near its target value, while in DrQ-v2 (pixel-based tasks), exploration is governed by scheduled exploration noise. Empirically, we confirmed that policy entropy or exploration noise remains stable regardless of sampling strategy, as shown in Table 13 of revised Appendix D.1. ####
>
> #### **Table 13.** Actor entropy statistics for three sampling strategies on Meta-World Sweep Into (10,000/50) with ten random seeds. ####
> |  | DS | NOS | PLS |
> |---|------|------|------|
> | Mean | -3.688 | -3.759 | -3.758 |
> | Std | 1.138 | 0.948 | 0.945 |
> | Max | 2.236 | 2.209 | 2.267 |
> | Min | -4.271 | -4.158 | -4.135 |
>
> ---
>
> >**Q2. Is a diversity constraint necessary?**
>
> #### We appreciate this insightful feedback. We believe an explicit diversity constraint is not necessary. Our design prioritized query-policy relevancy: by training the reward model on policy-aligned queries, it achieves accurate predictions in regions the policy frequently visits. This focus has proven effective across our experiments. Moreover, PLS already provides a mechanism to adjust this when needed through its temperature parameter $\alpha$ (Equation 4). Nevertheless, building on the reviewer's suggestion, incorporating explicit diversity constraints (similar to adaptive entropy tuning in SAC) or combining PLS with exploration methods like PPE [1] represent promising directions for achieving more refined balance between relevancy and diversity. ####
>
> ---
> #### **Reference** ####
> #### [1] Zhu, Y., Liu, J., Yuan, Y., Wei, W., Ge, Z., Fang, Z., ... & An, B. Optimizing Reward Models with Proximal Policy Exploration in Preference-Based Reinforcement Learning. In NeurIPS 2024 Workshop on Behavioral Machine Learning. ####

---

> > ### Comment · Reviewer_vBkx · 2025-11-25
> >
> > Thank you for the detailed rebuttal and the corresponding revisions.
> >
> > Most of my earlier concerns have been satisfactorily addressed:
> >
> > - The additional discussion of PPE and the clarification of how it relates to PLS (exploration for data collection vs. query selection from the replay buffer) make the positioning of this work in the literature much clearer.
> >
> > - The clarification around the effect of PLS on policy entropy, together with the empirical statistics reported in the revised appendix, helps reassure me that PLS does not systematically bias the underlying RL policy toward a low-entropy regime.
> >
> > - The discussion on query diversity and the role of the temperature parameter in PLS is helpful.
> >
> > Overall, I find the method well motivated and empirically convincing after the rebuttal. I therefore keep my overall score and recommendation for acceptance.

---

> > > ### Author Response · Authors · 2025-11-27
> > > **Thank you for your recommendation and thoughtful review!**
> > >
> > > Thank you for your thoughtful follow-up and for taking the time to review our revisions. Your comments significantly improved the clarity and presentation of our work, and we truly appreciate your constructive guidance throughout the process. We are grateful for your positive recommendation.

---

### Official Review · Reviewer_i7YB · 2025-11-02

**Soundness:** 2
**Presentation:** 3
**Contribution:** 2
**Rating:** 6
**Confidence:** 3

**Summary:**

This paper proposes PoLiCER, which introduces two mechanisms to improve preference-based reinforcement learning. 1) Policy Likelihood-based Sampling (PLS) selects feedback queries most aligned with the current policy. 2) Critic-Exploited Reset (CER) prevents reward overestimation from primacy bias by adaptively resetting the reward and critic networks. The goal of PLS is to ensure feedback queries remain representative of the current policy, avoiding outdated or irrelevant samples. Compared to recency-based methods, it directly measures alignment between data and current policy. It is more computationally efficient than disagreement sampling, requiring only $2 \times L \times K$ forward passes than $N$ for disagreement sampling. And it does not increase training cost. The goal of CER is to counteract primacy bias, where the reward estimator overfits to early feedback and inflates Q-values, leading to overoptimistic policies. It dynamically stabilizes reward learning by resetting networks only when critic overestimation is detected. Experiments are conducted on Meta-World and DMControl tasks. Authors compared to several existing baselines and show improvements.

**Strengths:**

PoLiCER offers several strengths over prior preference-based reinforcement learning methods such as disagreement sampling and recency-based query selection. Its Policy Likelihood-based Sampling (PLS) improves data–policy alignment by measuring how representative each trajectory is under the current policy, rather than assuming recency implies relevance. This allows the model to select feedback that directly reflects its current behavior, improving sample efficiency and policy convergence.

Another advantage is computational efficiency. Unlike ensemble-based disagreement sampling, which requires multiple forward passes per query, PLS operates with a fixed, small number of policy evaluations. Its inverse-rank likelihood weighting also provides robustness to outliers, enabling stable performance across diverse continuous control tasks. Together, these factors make PoLiCER both efficient and reliable in selecting informative feedback.

PoLiCER’s second component, Critic-Exploited Reset (CER), effectively mitigates reward overestimation caused by primacy bias. Instead of using fixed reset intervals, CER dynamically resets the reward estimator and critic only when critic outputs exceed an adaptive threshold. This approach reduces overestimation while allowing normal learning to continue when stable, leading to better long-term returns and less training disruption.

Overall the writing is clear and easy to follow. Experimental setups are well explained, together with baseline methods. In the DMControl suite, it achieved performance comparable to QPA and clearly outperformed earlier PbRL methods like PEBBLE, SURF, RUNE, and MRN. In the more challenging Meta-World benchmarks, PoLiCER further distinguished itself by achieving over 80% average success, significantly higher and more consistent than competing methods.

**Weaknesses:**

1. PLS depends on accurate policy likelihood estimation, which may be unreliable in highly stochastic or multimodal policies.
2. The rank-based weighting, though robust, can blur distinctions between highly relevant and moderately relevant samples.
3. CER’s adaptive resets introduce temporary instability as networks reinitialize, requiring careful tuning of replay ratios.
4. Because PoLiCER omits ensemble-based uncertainty modeling, it may handle noisy or inconsistent human feedback less effectively than Bayesian or disagreement-based methods.

**Questions:**

It seems that PoLiCER has more empirical performance gain in pixel-based environments, is there any intuition on why this happens compared to state-based environments?

---

> ### Author Response · Authors · 2025-11-22
> **Response to Reviewer i7YB (Part1/2)**
>
> We sincerely appreciate the reviewer’s insightful comments and constructive feedback. We believe that addressing the raised concerns has significantly improved the quality of our paper. We have carefully revised the paper to address based on your suggestions, with all changes highlighted in blue for easy identification. We respond to each comment as below.
>
> ---
>
> > **W1. PLS depends on accurate policy likelihood estimation, which may be unreliable in highly stochastic or multi-modal policies.**
>
> #### We appreciate this important concern. While stochasticity and multi-modality are inherent challenges in RL that affect all likelihood-based methods, our design specifically addresses their impact on query selection through rank-based weighting. By using relative rankings rather than absolute likelihood values, we mitigate estimation noise and outlier sensitivity in highly stochastic or multi-modal policies. This ensures stable query selection even when likelihood magnitudes are uncertain. Our rank-based approach thus provides robustness against the very concerns raised by the reviewer, making PLS reliable across diverse policy characteristics. ####
>
> ---
>
> >**W2. The rank-based weighting, though robust, can blur distinctions between highly relevant and moderately relevant samples.**
>
> #### Thank you for pointing out this. We agree that rank-based weighting may compress distinctions between highly and moderately relevant samples. Therefore, we use a temperature parameter $\alpha$ (Equation 4) to preserve fine-grained distinction between samples by controlling the sharpness of the sampling distribution over the already-stabilized ranks. This design combines the robustness of rank-based normalization against outliers and estimation errors with the flexibility to recover detailed distinctions through temperature control, achieving both stability and informativeness in query selection. ####
>
> ---
>
> >**W3. CER’s adaptive resets introduces temporary instability as networks reinitialize, requiring careful tuning of replay ratios.**
>
> #### We acknowledge that resets can introduce temporary instability. However, as discussed in Section 2, reset-based methods remain the most effective approach for addressing plasticity loss [1], and our experimental results demonstrate that CER’s benefits far outweigh this temporary instability. Importantly, replay ratio tuning is straightforward rather than requiring careful manual adjustment. Following empirical findings that low replay ratios preserve plasticity in early training while high ratios improve later-stage sample efficiency [2,3], we simply increase the replay ratio by 1 after each reset (triggered only when overestimation is detected). This minimal adaptive schedule effectively balances plasticity preservation with data efficiency across all tasks, without requiring task-specific hyperparameter tuning. ####
>
> ---
> #### **Reference** ####
>
> #### [1] Nauman, M., Bortkiewicz, M., Miłoś, P., Trzciński, T., Ostaszewski, M., & Cygan, M. (2024, July). Overestimation, overfitting, and plasticity in actor-critic: the bitter lesson of reinforcement learning. In Proceedings of the 41st International Conference on Machine Learning (pp. 37342-37364). ####
> #### [2] Ma, G., Li, L., Zhang, S., Liu, Z., Wang, Z., Chen, Y., ... & Tao, D. Revisiting Plasticity in Visual Reinforcement Learning: Data, Modules and Training Stages. In The Twelfth International Conference on Learning Representations. ####
> #### [3] D'Oro, P., Schwarzer, M., Nikishin, E., Bacon, P. L., Bellemare, M. G., & Courville, A. Sample-Efficient Reinforcement Learning by Breaking the Replay Ratio Barrier. In The Eleventh International Conference on Learning Representations. ####

---

> ### Author Response · Authors · 2025-11-22
> **Response to Reviewer i7YB (Part2/2)**
>
> > #### **W4. Because PoLiCER omits ensemble-based uncertainty modeling, it may handle noisy or inconsistent human feedback less effectively than Bayesian or disagreement-based methods.** ####
>
> #### We respectfully clarify that query sampling strategy and handling noisy feedback are distinct concerns in PbRL. Disagreement-based sampling (DS) selects queries based on ensemble variance, measuring model uncertainty about the reward function for query informativeness rather than reliability or consistency of human annotations as described in revised Appendix C.1. All PbRL methods are equally vulnerable to label noise regardless of query sampling strategy. The key to handling such noise lies in the robust loss function or noisy data filtering [4,5] rather than query selection. ####
>
> #### Importantly, PoLiCER demonstrated strong performance in actual human-in-the-loop experiments involving inherently noisy and inconsistent feedback from diverse annotators (revised Section 5.5, now including Window Open; five runs currently, extending to ten in the final version). This robustness stems from CER’s periodic resets, which prevent the reward model from overfitting to early, potentially unreliable preferences and disrupt primacy bias cycle. ####
>
> ---
>
> > #### **Q1. It seems that PoLiCER has more empirical performance gain in pixel-based environments, is there any intuition on why this happens compared to state-based environments?** ####
>
> #### We appreciate the reviewer's insightful observation. To better understand this phenomenon, we examined recent literature on learning dynamics in pixel-based RL. Prior work has shown that pixel-based RL suffers from severe plasticity loss, often manifesting as dormant neurons [6, 7]. This issue is particularly pronounced in pixel-based settings compared to state-based environments [8]. Since network resets effectively mitigate this degradation, our proposed CER exhibits stronger effects in pixel-based tasks, leading to the larger performance gains observed. ####
> ---
>
> #### **Reference** ####
> #### [4] Cheng, J., Xiong, G., Dai, X., Miao, Q., Lv, Y., & Wang, F. Y. (2024, July). RIME: Robust Preference-based Reinforcement Learning with Noisy Preferences. In International Conference on Machine Learning (pp. 8229-8247). PMLR. ####
> #### [5] Huang, S., Levy, M., Gupta, A., Ekpo, D., Zheng, R., & Shrivastava, A. (2025). TREND: Tri-teaching for robust preference-based reinforcement learning with demonstrations. In Proceedings of the IEEE International Conference on Robotics and Automation (ICRA) ####
> #### [6] Sokar, G., Agarwal, R., Castro, P. S., & Evci, U. (2023, July). The dormant neuron phenomenon in deep reinforcement learning. In International Conference on Machine Learning (pp. 32145-32168). PMLR. ####
> #### [7] Xu, G., Zheng, R., Liang, Y., Wang, X., Yuan, Z., Ji, T., ... & Xu, H. DrM: Mastering Visual Reinforcement Learning through Dormant Ratio Minimization. In The Twelfth International Conference on Learning Representations. ####
> #### [8] Ji, T., Liang, Y., Zeng, Y., Luo, Y., Xu, G., Guo, J., ... & Xu, H. ACE: Off-Policy Actor-Critic with Causality-Aware Entropy Regularization. In Forty-first International Conference on Machine Learning. ####

---

> ### Author Response · Authors · 2025-11-28
> **Looking forward to your response**
>
> Dear reviewer i7YB,
>
> We sincerely appreciate your detailed and insightful review. As the discussion period draws to a close, we wanted to kindly follow up to see if our responses and revisions have adequately addressed your concerns. We have carefully responded to each point you raised, and we would be grateful to know if any questions remain that we could clarify during this remaining time. We fully understand the demands on your time and deeply value your feedback, which has been invaluable for improving our work. Thank you again for your contribution to this review process.
>
> Best regards,
>
> Authors

---

### Author Response · Authors · 2025-12-01

### **Summary of Discussion and Revisions** ###

We thank all reviewers for their thoughtful insights and constructive feedback on our work. We have carefully addressed each concern and updated our paper to reflect the suggestions. Newly added and updated sections are highlighted in blue. We briefly summarize the key revisions below:

---
**Reviewer i7YB**: We addressed concerns about PLS robustness in stochastic/multi-modal policies by clarifying that rank-based weighting mitigates outliers while temperature parameter $\alpha$ enables fine-grained control. We clarified that CER requires no task-specific tuning, and distinguished that Disagreement sampling (DS) measures model uncertainty rather than feedback reliability (in `Appendix C.1`). We also explained that PoLiCER shows stronger gains in pixel-based environments due to more severe plasticity loss compared to vector-based environments.

---
**Reviewer vBkx**: We added discussion of PPE in `Section 2` and empirical entropy statistics in `Table 13 (Appendix D.1)`, demonstrating PLS does not bias toward low-entropy policies. We also clarified that temperature parameter $\alpha$ provides sufficient control over query diversity.

---
**Reviewer SQmv**: We expanded HiL experiments to include the Window Open environment. Comprehensive results are detailed in `Section 5.5`, with procedures and an analysis of mixed preferences provided in `Appendix D.6`. We added computational analysis in `Table 15 (Appendix D.5)`, showing training time overhead across various PbRL algorithms for pixel-based control. We clarified with ablations in `Section 5.3` that performance gains mainly stem from PLS + CER rather than TA.

---

### Meta-Review · Area_Chair_3VSj · 2025-12-22

**Summary:**

The reviewers appreciated the combination of (1) a method for selecting on-policy trajectories to generate more relevant queries in RLHF and critic network reset to avoid overestimation. However, they raised various concerns about the need of accurate policy likelihood estimation, blurring effect due to rank-based weighting, instability induced by network reset, handling of noisy or incorrect human feedback, sensitivity to policy entropy, missing related work, lack of theoretical foundation, simplicity of human-in-the-loop experiments, or difference with SURF.

**Reviewer Concerns:**

I believe that all the raised concerns were well-addressed apart from missing related work and the lack of theoretical foundation. While I understand the lack of theoretical foundation for this kind of experimental work, I think the authors should have done a better work discussing the rich literature in preference-based RL, active reward learning, and more recent work in RL from human feedback (RLHF). In particular, I find it a bit peculiar that the authors never mentioned RLHF. I believe that it is the more modern accepted term for PbRL, which includes older work with a different emphasis (e.g., see surveys: Wirth et al., JMLR 2017 and Kaufmann et al., TMLR 2024). The authors can check the latter survey for previous work that focuses on on-policy trajectories and active reward learning. In addition, their proposed policy likelihood is also related to a previous proposition (Feng et al., AAAI 2025).

**Reviewer Scores:**

I believe that the reviewers would have kept their scores.

---

### Decision · Program_Chairs · 2026-01-26

Accept (Poster)